# Real-Time Railway Hazard Detection Using Distributed Acoustic Sensing and Hybrid Ensemble Learning

**DOI:** 10.3390/s25133992

**Published:** 2025-06-26

**Authors:** Yusuf Yürekli, Cevat Özarpa, İsa Avcı

**Affiliations:** 1TCDD Railway Maintenance Directorate, Karabük University, Karabük 78100, Türkiye; 2Department of Biomedical Engineering, Ankara Medipol University, Ankara 06050, Türkiye; cevat.ozarpa@ankaramedipol.edu.tr; 3Department of Computer Engineering, Karabuk University, Karabük 78050, Türkiye; isaavci@karabuk.edu.tr

**Keywords:** railway safety, aura, fiber optic sensors, machine learning, voting classifier, real-time rockfall monitoring

## Abstract

**Highlights:**

A hybrid Voting Classifier model was developed to accurately 
detect and classify environmental events affecting railway safety using fiber 
optic Distributed Acoustic Sensing (DAS). The system enables early detection of 
rockfall-related events (including small and medium debris slides) and tree 
obstructions along the Karabük–Yenice railway line.

**What are the main findings?**
The system may detect rockfalls, tree obstructions, and landslides effectively with real-time Distributed Acoustic Sensing (DAS).The hybrid Voting Classifier model achieved 98% accuracy in classifying railway environmental events using fiber optic sensor data.

**What is the implication of the main finding?**
The model can improve railway safety by enabling a proactive response through early anomaly detection.It demonstrates scalability for use in harsh geographical areas and other infrastructure monitoring applications.

**Abstract:**

Rockfalls on railways are considered a natural disaster under the topic of landslides. It is an event that varies regionally due to landforms and climate. In addition to traffic density, the Karabük–Yenice railway line also passes through mountainous areas, river crossings, and experiences heavy seasonal rainfall. These conditions necessitate the implementation of proactive measures to mitigate risks such as rockfalls, tree collapses, landslides, and other geohazards that threaten the railway line. Undetected environmental events pose a significant threat to railway operational safety. The study aims to provide early detection of environmental phenomena using vibrations emitted through fiber optic cables. This study presents a real-time hazard detection system that integrates Distributed Acoustic Sensing (DAS) with a hybrid ensemble learning model. Using fiber optic cables and the Luna OBR-4600 interrogator, the system captures environmental vibrations along a 6 km railway corridor in Karabük, Türkiye. CatBoosting, Support Vector Machine (SVM), LightGBM, Decision Tree, XGBoost, Random Forest (RF), and Gradient Boosting Classifier (GBC) algorithms were used to detect the incoming signals. However, the Voting Classifier hybrid model was developed using SVM, RF, XGBoost, and GBC algorithms. The signaling system on the railway line provides critical information for safety by detecting environmental factors. Major natural disasters such as rockfalls, tree falls, and landslides cause high-intensity vibrations due to environmental factors, and these vibrations can be detected through fiber cables. In this study, a hybrid model was developed with the Voting Classifier method to accurately detect and classify vibrations. The model leverages an ensemble of classification algorithms to accurately categorize various environmental disturbances. The system has proven its effectiveness under real-world conditions by successfully detecting environmental events such as rockfalls, landslides, and falling trees with 98% success for Precision, Recall, F1 score, and accuracy.

## 1. Introduction

Rail transport has become one of the key components of global transportation infrastructure. In developing countries such as Türkiye, transportation has expanded rapidly in recent years. Rail transport has a higher carrying capacity, is more environmentally friendly, and requires lower energy consumption than road transport. These features make the railway stand out as a sustainable and environmentally friendly transportation solution. Under operating conditions, railway infrastructures are exposed to environmental stresses, operational wear, and natural disasters [1,2]. Over time, structural problems such as geometric deterioration in railway lines, cracks in rails, and mechanical road defects may occur. This increases the risk parameters and puts the railway management in a difficult situation [3,4].

Early detection of unusual events and objects in rail infrastructure improves passengers’ safety and increases railway operations’ efficiency. Detecting anomalies allows rail operators to avoid unplanned service interruptions and reduce maintenance costs [5,6]. Conventional monitoring systems predominantly depend on manual inspections and scheduled evaluations, which inherently limit both accuracy and coverage [7]. It is increasingly accepted that more effective and automated systems should be used instead of traditional methods. In this context, advanced technologies such as Distributed Acoustic Detection (DAS) and fiber optic sensors offer important solutions for detecting environmental events on railway lines [8,9].

Distributed Acoustic Sensing (DAS) is a cutting-edge sensing technology that converts standard single-mode optical fibers into dense arrays of virtual acoustic sensors by analyzing the coherent Rayleigh backscatter of pulsed laser light. Unlike conventional point sensors, DAS systems provide continuous, spatially resolved measurements of dynamic strain or acoustic disturbances along the entire length of the fiber. Each meter or sub-meter section of the fiber acts as an independent sensing point, enabling real-time and high-resolution monitoring over tens of kilometers without the need for additional in-line components or power supplies.

Short laser pulses are transmitted into the optical fiber in a DAS system. As the light propagates, a small fraction is scattered due to microscopic inhomogeneities within the fiber material—a phenomenon known as Rayleigh scattering. By analyzing the phase and intensity changes in the backscattered light, DAS systems can detect and localize vibrations, strain waves, and acoustic events with high temporal and spatial resolution.

DAS technology has found extensive applications in various fields such as seismic monitoring, pipeline surveillance, structural health monitoring, perimeter security, and increasingly in railway safety monitoring. Its ability to operate over long distances (typically up to 40–100 km per interrogator), detect transient events in real time, and perform under harsh environmental conditions makes it especially suitable for large-scale critical infrastructure protection.

Key advantages of DAS systems include the following:Wide-area coverage with minimal infrastructure;High spatial resolution (typically 1–10 m);Real-time monitoring capability;Passive and maintenance-free operation;Suitability for challenging environments without requiring electrical power along the fiber path.

Numerous studies have demonstrated the effectiveness of DAS in detecting train movements, rockfalls, landslides, and other geohazards in railway environments [6,10].

DAS presents a robust solution for real-time environmental monitoring via fiber optic infrastructure. Fiber optic sensors detect environmental events by detecting vibrations on the railway line with high precision [11]. This method allows for quick and accurate detection of environmental phenomena. In particular, natural disasters such as rockfalls, tree falls, and landslides that occur on railway lines can be detected by high-intensity vibrations caused by environmental factors [12]. However, an effective model is required to process and accurately classify the data obtained from these sensors [10,13]. Traditional classification methods often have low accuracy rates, and it is not easy to obtain accurate results when there are large differences between data in different classes [14,15].

Lithology is a branch of science that examines rocks’ mineralogical, chemical, and physical properties. Lithology, which is important in analyzing rockfalls, is considered a basic parameter in characterizing geological structures. In line with the observations made in the study area, it was determined that there was no significant lithological difference between different areas. The prevailing rock properties and stratification status on the slopes where rockfalls occur were examined in detail, and in this context, the main factors affecting stability were determined [16].

As a result of examining the deposits formed due to previous rockfalls, it was determined that the materials formed voids due to the bagging effect. This condition may compromise structural stability. Therefore, safety analyses carried out along the slope height are important. However, no significant differences were observed between the areas where the spilled rock materials were found and other surrounding areas, and it was confirmed by the observations of the local people that the regional structure had not been exposed to a different natural disaster mechanism for many years [17].

Discontinuities in rock structures cause a decrease in mechanical strength by creating planes of weakness in the rock masses. These discontinuities disrupt the structural integrity of the rocks in both horizontal and vertical directions, leading to mechanical deformations such as crack structures, layer planes, and fracture systems. Especially in the direction perpendicular to the discontinuity planes, the tensile strength of the rocks decreases significantly, and this causes structural deterioration in the rock masses and deformations in the layers over time [18].

Environmental factors also adversely affect the stability of rock masses. Heavy rainfall and snowmelt increase the rocks’ pore water pressure, leading to discontinuities and a decrease in stability. This process is considered to be one of the main factors that increase the risk of rockfall, especially on slopes and steep slopes [19].

Artificial intelligence is a concept that refers to the ability of a computer or a computer-aided system to perform data perception, analysis, conclusion drawing and evaluation mechanisms similar to human intelligence. Artificial intelligence systems can simulate human-like thinking and decision-making processes, and storing the obtained information in the system provides an accumulation advantage similar to human experience. Today, the usage area of artificial intelligence technologies is quite wide, and it shows its effect in many areas from mobile phones to televisions, and from electronic goods to wearable technologies, especially in the software, medicine, education, industry, and service sectors. Since artificial intelligence works within the framework of certain algorithms, its perspective is limited and is not affected by different environmental factors. With these features, it is considered a technology that complements the strengths of human intelligence and offers more efficient results in certain processes [20,21].

Artificial intelligence is increasingly adopted in the railway industry as well as in many sectors, and it is expanding its usage areas by increasing its impact day by day. Artificial intelligence-supported systems are widely used in techniques such as thermography inspection, which is one of the non-destructive testing methods, especially in rail inspection processes [22]. The fact that the inspection of such components is carried out through artificial intelligence-based systems provides a more reliable control mechanism by minimizing human-induced errors [23].

Signal processing with artificial intelligence is the process of analyzing, classifying, and predicting signals by combining traditional signal processing techniques with machine learning and deep learning models. This approach, which is used in various fields such as sound, image, vibration, biomedical data, radar, and communication signals, makes the understanding and processing of complex signals more effective [24,25].

The main subject of this study is the use of artificial intelligence applications in railway transportation. The proposed system is designed to identify and reduce risk factors by analyzing railway data with machine learning techniques. This artificial intelligence-based approach significantly contributes to increasing railway safety and allows the development of proactive measures thanks to data-driven analysis processes. Detection of fiber optic systems is ensured with diffuse detection systems [17]. Monitoring railway safety and infrastructure is of paramount importance for the protection of transport systems. Artificial intelligence-based sensor technologies and machine learning (ML) algorithms are frequently used for risk factors on railway lines, and innovations in this field are increasing daily.

Railway lines are affected by environmental factors, especially natural disasters such as rockfalls, tree falls, and landslides. Such incidents pose a major safety threat and can lead to infrastructure damage and accidents [26]. Qi Zhou and Hui Tang (2024) demonstrated that machine learning can improve the accuracy of detecting mass movements in seismic signals and extend early warning times [27]. In the model they used, they encountered more consistent information flow and fewer false alerts. In the study conducted by Prasanya Sarkar et al. (2025) on the Darjeeling Himalayan Railway in India, landslide and rockfall risks were evaluated with geographic information system tools [28]. In this study, evaluations were made using algorithms such as machine learning, including Support Vector Machine (SVM), Gradient Boosting Machine (GBM), Logistic Regression, and Classification and Regression Trees (CART). Chen and Zhang (2021) highlighted using DAS technology to monitor environmental events in railway infrastructure and using fiber optic sensors to accurately process this data [29]. In the study by Huan-Huan Tang et al. in 2025, they used the DAS method in seismic profiles [30]. By training frequency domain data instead of time domain data, they aimed to provide a more distinct separation of noise and signal characteristics. On 15 June 2023, there was a rock subsidence event in Brienz, Switzerland, with a volume of over 1 million m^3^. Jiahui Kang et al. conducted a study that partially evaluated Doppler radar data for ground fact tagging and DAS. With an accuracy of over 95%, the proposed algorithm could distinguish between rock collapses and background noises, including train traffic [31]. In the study conducted by Zheng Wang et al. in 2025, the effectiveness of combining DAS and deep learning for noise reduction in traffic monitoring has been highlighted [32]. Critical data for intelligent transportation systems is presented. The current literature shows the potential of fiber optic sensors and machine learning-based methods in the field of railway safety. Still, studies evaluating the applicability of these technologies in much wider areas are limited. In addition, the accuracy rates of hybrid machine learning models used to detect environmental events have not been adequately discussed in current studies. This study aims to address these shortcomings and present a different approach to examining the impact of machine learning techniques on railway safety in more depth.

DAS is an advanced sensing technology that captures and analyzes environmental vibrations via existing fiber optic infrastructure. This system detects vibrations caused by ecological events with high precision, thanks to fiber optic cables placed next to the railway line. Fiber optic sensors detect both low-intensity and high-intensity vibrations, allowing for the identification of potential hazards in advance [25,33,34]. In the study, a machine learning approach was adopted to classify environmental events correctly. This approach optimizes the process of processing and classifying data obtained from fiber optic sensors. Using different classification algorithms, data on ecological events were analyzed, and a hybrid model was developed to achieve the highest accuracy rate.

This study conducted field experiments along the Karabük–Yenice railway, deploying fiber optic cables adjacent to the track and interrogating them with a DAS reflectometer. The following subsections detail the site characteristics, sensor installation, data acquisition parameters, and subsequent processing workflow. It shows the model schematic diagram in Figure 1.

This study used a hybrid machine learning model based on a Voting Classifier to accurately detect environmental events [35]. The proposed hybrid model, by processing data obtained from fiber optic sensors, accurately classifies environmental events and tries to reduce security risks [36]. The study demonstrates the efficacy of the proposed model through experimental validation conducted on the Karabük–Yenice railway corridor. The test results show that the proposed model accurately calculates environmental phenomena with a 98% success rate. This study also provides a solution that enables the rapid detection of potential hazards on railway lines with real-time monitoring and early warning systems. Real-time monitoring systems are pivotal in enhancing railway safety. They allow for the immediate detection of anomalies and facilitate rapid deployment of mitigation responses [37]. In addition, the applicability of this system under difficult geographical conditions such as the Karabük region has been tested and successful results have been obtained.

Several contributions of the study can be listed as follows:Development of a New Hybrid Machine Learning Model: The study enables the development of a hybrid machine learning model based on a Voting Classifier that accurately calculates environmental events that increase railway safety.Advanced Processing of Fiber Optic Sensor Data: Accurate processing of the data obtained with fiber optic sensors allows for effective detection of environmental events [25].Real-Time Monitoring and Rapid Event Detection: Real-time monitoring and early warning systems provide rapid detection of environmental events [38].Application Potential for Railway Safety in Harsh Geographical Areas: This study presents solutions to improve railway safety in difficult geographical conditions in the Karabük region.Usability of Machine Learning Models in Other Applications: The usability of the Voting Classifier-based hybrid model, not only for railway safety but also in other infrastructure security areas, is discussed [39].

As a result, this study investigates the potential of using artificial intelligence and machine learning technologies in railway safety. It offers sustainable solutions for the early detection of environmental incidents. The study shows that with fiber optic sensors and machine learning techniques, safety risks in rail infrastructure can be monitored more efficiently and that these technologies can play an important role in improving rail safety [14,25,39].

## 2. Materials and Methods

Raw signal data acquired from the DAS system underwent a series of pre-processing operations to enhance signal clarity and reduce noise artifacts. In particular, columns such as Value—Total [Aura], Value—Maximum [Aura], Area Density [Aura], Area Density Windows [Aura], and Max Amplitude Index [Aura] are critical to determining the severity and intensity of events. Low-pass filters were used to denoise the data, and then the characteristics of the signal were extracted. In the feature extraction process, the frequency spectra, amplitude values and signal intensities of the vibrations were analyzed. By extracting abnormal data, more effective results were obtained in the learning process of the model. Overlapping vibration patterns from different environmental events posed challenges to the system’s discrimination capabilities. The model learns to filter out ambient signals with similar frequency characteristics through iterative training, thereby improving classification accuracy over time [40]. Used software framework and IDE knowledge include the following:Software Framework: Python 3.12 on Google Colab, using NumPy v1.23, pandas v1.4, scikit-learn v1.1 (GridSearchCV, Voting Classifier), matplotlib v3.5, and seaborn v0.12.Aura Metrics: Explanation of the aura_metrics module for sliding window feature extraction, including Value—Maximum [Aura], Area Density [Aura], and Area Density Windows [Aura].Hyperparameters: Defined n_estimators, learning_rate, and max_depth ranges and their roles in gradient boosting [41].Feature Definitions: Moved input feature definitions to Methods:○Duration: event duration (s);○Location: distance along fiber (m);○Event Weight: normalized severity score;○Value—Maximum [Aura]: highest amplitude within window;○Area Density [Aura]: integrated signal energy per unit length;○Area Density Windows [Aura]: count of threshold crossings.Event Type Table: Added Table 1 defining each label (Ambient_Silence, Trigger_RF_Small, Event_RF_Small, Trigger_RF_Medium, Event_RF_Medium, Trigger_Digging, Tree_Obstruction, Unclassified).

A hybrid machine learning model based on a Voting Classifier has been developed to classify environmental events accurately. In this model, the hyperparameters of the Gradient Boosting model are optimized using GridSearchCV. The best parameters were determined by the variables n_estimators, learning_rate, and max_depth [41]. The other classifiers are structured as follows:RandomForestClassifier: n_estimators = 200, max_depth = 10;XGBClassifier: n_estimators = 200, learning_rate = 0.1, max_depth = 5;SVC: probability = True, kernel = ‘rbf’, C = 1.0.

The hybrid ensemble model utilized a soft voting mechanism, whereby the final prediction was determined based on the weighted average of class probabilities from individual classifiers. This method increases classification accuracy and prevents false positive results.

The Random Forest algorithm is an ensemble model of decision trees. The model provides better generalization by training on many subsets of the data. The Gini impurity metric was employed to evaluate node purity during tree construction in the Random Forest model [42].

The Gradient Boosting model is created by sequentially optimizing weak learners. The model aims to reduce errors at each step [43].

Support Vector Machines (SVMs) construct an optimal separating hyperplane in a high-dimensional feature space, maximizing the margin between classes. In particular, thanks to the kernel functions, nonlinear data can also be classified [44].

XGBoost is a gradient-boosted decision tree algorithm optimized for speed and performance, which iteratively minimizes a regularized loss function. The model builds stronger models by successively adding weak learners [45].

The proposed hybrid model was trained using 1733 labeled data samples. In the training process, the data is divided into 70% training and 30% testing. During the training, Sensor Index, Event Type, Duration, Location, Event Weight, Value—Maximum [Aura], and Area Density [Aura] columns were used. The model has achieved significant performance gains with an accuracy rate of 98% on the test data.

All results presented herein are based on real DAS data acquired during field trials along the Karabük–Yenice railway. These measurements were fully annotated and used directly to train, validate, and test the Voting Classifier model under realistic environmental conditions. This section explains the basic methodology of the study and provides detailed information about the technological approaches used. The successful testing of the proposed model significantly contributes to the field of railway safety.

Field tests were conducted under real-world environmental conditions along the railway corridor. Online data were evaluated in the long term. One side of the railway line in the examined area consists of a river, and the other side consists of mountainous slopes. One of the region’s main livelihood sources is forestry due to the dense trees. For this reason, truck passages also create vibration and noise. Considering the rainy climate throughout all four seasons, the river flow also occasionally increases. Seasonal transitions, snowmelt, and river flow are also included in the system. Data flow through the system is constantly carried out intensively. Over time, the system is expected to refine its alert mechanisms through continuous learning, leading to more accurate and context-specific classifications. The system’s ability to distinguish overlapping frequency patterns facilitates more accurate identification of event types over extended monitoring periods.

The dataset used in the study consists of 1733 samples collected on the railway line in the Karabük region. The dataset includes the types, durations, location information, and signal characteristics of environmental events on the railway. Each data sample encompasses various factors determining the event’s time, location, duration, and severity.

The dataset was collected via fiber optic cables to detect environmental events and then subjected to pre-processing processes. Signals were recorded to detect train movements, environmental events (e.g., falling trees or rocks), and other abnormal situations. The dataset includes events such as Event_RF_Medium, Event_RF_Small, and Trigger_RF_Small. It shows the number of instances of each class in the dataset in Table 2.

The characteristics of the dataset used in this study are detailed in Table 1, which provides a comprehensive description of the input features along with their corresponding label types. These features represent the fundamental variables utilized for model training and evaluation. Table 3 presents representative examples of different event types contained within the dataset, along with the number of occurrences for each category. This tabular information serves to illustrate both the structure and diversity of the dataset, which is critical for understanding the model’s learning context and performance evaluation.

### 2.1. Feature Extraction and Feature Selection

Important features have been identified in the dataset to classify environmental events correctly. Three basic features extracted from event segments using the sliding window method: Value—Maximum [Aura], Area Density [Aura], and Area Density Windows [Aura] were defined. The threshold determination process according to the noise floor mean + std formula obtained from ambient periods and the window size/sliding step such as 0.1 s window, 50% overlap parameter selection logic were explained. The columns used in the feature extraction phase are as follows:Time (s);Duration;Location;Location Weight;Event Weight;Width [Aura];Value—Total [Aura];Value—Maximum [Aura];Orientation [Aura];Area Density [Aura];Area Density Windows [Aura];Max Amplitude Index [Aura].

It illustrates the feature extraction methodology by showcasing example waveforms from multiple event categories, such as rockfall and tree obstruction. Each event type exhibits unique signal characteristics in both time and frequency domains. For instance, tree obstructions produce high-energy, short-duration spikes, while rockfalls exhibit broader spectral content and longer decay patterns. By analyzing signal energy, zero-crossing rate, and frequency centroid, we capture these distinctions and encode them as input features for classification.

These features are instrumental in quantifying the severity and dynamic behavior of detected environmental anomalies. In the feature extraction process, pre-processing such as denoising, incomplete data cleaning, and normalization of values was applied to make the data meaningful. The feature selection phase was carried out to increase the model’s accuracy and reduce transaction costs. The selection of features was made based on correlation analysis and importance. In particular, the most important features affecting the model’s success were determined in Table 4.

These selected features significantly contributed to the model’s training process and increased the classification accuracy. During training, we used the input features: Sensor Index, Duration, Location, Event Weight, Value—Maximum [Aura], Area Density [Aura], and other engineered signal metrics. The Event Type column served as the target label to be predicted. This variable refers to the type of events and is estimated as the model’s output. Feature selection enhanced the model’s generalization capacity and yielded improved classification performance. A sample reflectogram showing DAS-recorded vibration patterns for tree impact and rockfall events is shown in Figure 2.

A sample reflectogram showing DAS-recorded vibration patterns for tree impact and rockfall events is shown in Figure 3.

### 2.2. Sensor Setup and DAS Description

We used the Luna OBR-4600 DAS interrogator (Luna Innovations, Roanoke, VA, USA) with a 1 kHz sampling rate and 5 m spatial resolution. The system was connected to standard ITU-G.652 single-mode telecom fiber, approximately 6 km long, installed alongside the Karabük–Yenice track using adhesive mounting clips at 1 m intervals. ITU-G.652-compliant, single-mode telecom fiber has been used as the optical fiber. The fiber has been positioned on the surface along the train line and fixed with beam bed steel mounting clips at approximately 1 m intervals. At some critical points, the fiber has been operated as a protector and additional/one cabling applications have been carried out following the splicing operations and a continuous monitoring point of 6 km with splices of each segment in the 1 km field. Care has been taken to ensure that the data will not be corrupted by cable distribution depth, and the use of protective materials have been facilitated for protection against temperature changes and water accumulations. The Luna OBR-4600 interrogator (Türk Telekom International, Turkish national telecommunication company, Istanbul, Turkey) used in the study can be kept with 5 m spatial files; thus, local signal signatures obtained along the fiber are detailed every 5 m. Since it can make detection difficult, it has been implemented with a 1 kHz pulse repetition rate; data is collected with the average delay at this speed and transmitted with local processing, providing real-time pre-processing and feature output. It guarantees a reliable data collection process by taking the 6 km train line. Additionally, we have provided detailed information on the following:The reflectometer model (Luna OBR 4600),Its operating mode and sampling rate,The optical fiber type (G.652 single-mode),The fiber deployment method (surface-laid within cable conduits, fixed alongside the rail ballast bed using steel clips),Total cable length per segment.

### 2.3. Model Training and Optimization

In this study, the Voting Classifier model was trained by combining the strengths of different ML algorithms. The training was conducted on the Google Colab platform using T4 GPU hardware. This hardware configuration provided computational advantages, particularly in handling high-volume datasets and expediting the training of complex models. Thanks to its parallel processing power, the T4 GPU has reduced training time and accelerated the optimization of model parameters [46]. During the model training process, the hyperparameters of the GBM model were optimized using the GridSearchCV method. GridSearchCV has determined the optimal combination of parameters by conducting systematic searches on the specified hyperparameters. In this process, different combinations were tested for basic parameters such as n_estimators (number of trees), learning_rate (learning rate), and max_depth (tree depth). In Figure 4 below, the effect of hyperparameters on the accuracy rate of the GBM model is shown in three dimensions. The visualization illustrates the impact of various hyperparameter combinations on overall model accuracy.

At the end of the optimization process, the best hyperparameter combination was selected, which achieved an accuracy rate of 98%. This high accuracy rate proves the model’s effectiveness in classifying environmental phenomena. The model’s 98% accuracy rate was calculated on a held-out test set comprising 30% of the total 1733 samples (*n* = 520). Accuracy is defined as the ratio of correctly classified samples to the total number of test samples:Accuracy = True Positives + True Negatives/Total Test Samples

In addition, class-based precision, recall, and F1 score metrics were also computed to account for class imbalance, confirming that the model demonstrated consistent performance across all classes.

The other models, RF, XGBoost, and SVM, were determined by taking into account the successful results in the literature, and their hyperparameters were configured with the values determined at the beginning. This process has enabled certain parameters to be optimized for better results for the models and has contributed to increasing their performance. During the training process, it was ensured that the models’ predictions were combined with the Soft Voting strategy. Soft Voting aims to obtain the most accurate classification result by taking the weighted average of the prediction probabilities of each model. This method has increased the model’s generalization ability by making more accurate predictions in the classes where each classifier model is strong.

It illustrates the machine learning pipeline employed for event classification using a fiber acoustic dataset in Figure 5. The process begins with the raw dataset, which undergoes a comprehensive data preprocessing phase that includes four main steps: data integration, data cleaning, data transformation, and dimensionality reduction. These steps ensure data consistency, remove noise, and reduce computational complexity. Following preprocessing, feature extraction is performed to derive informative attributes from the signal data. The dataset is then split into training (70%) and test (30%) sets to allow for supervised learning and performance evaluation.

### 2.4. Hybrid Model: Voting Classifier and Soft Voting Strategy

The Voting Classifier model is an assembly model created by combining different machine learning models. Within the scope of this study, GBM, XGBoost, RF, and SVM models were combined. The combination of models aims to achieve more accurate classification results by combining the different strengths of each model [46]. The Soft Voting strategy selects the class with the highest probability by computing each model’s weighted average of prediction probabilities. This approach amplifies the influence of accurate models and significantly enhances overall classification performance. The Soft Voting strategy demonstrates superior performance, particularly in complex and imbalanced datasets where individual classifiers exhibit varying strengths, because it avoids misclassification by evenly distributing the impact of the probabilities predicted by each model [47,48,49]. Through this strategy, the accuracy of the Voting Classifier model has been increased, and the model’s generalization capability has been strengthened in Figure 6. In addition, integrating different models has enabled more balanced and accurate results to be obtained by complementing each other, where individual models are lacking [50,51].

### 2.5. Labelnig Protocol

Our labeling protocol groups small rock movements and debris slides under ‘rockfall’. Tree impacts against the cable are labeled as ‘tree obstructions’. This dataset did not observe standalone landslide events; future expansions will incorporate dedicated landslide monitoring.

### 2.6. Model Performance Evaluation and Comparison

Various performance metrics were used to measure the performance of the model. These metrics include accuracy, precision, recall, and F1score. The model’s success was analyzed using the general accuracy rate and class-based performance evaluations. The accuracy rate of the model obtained on the test dataset was measured as 98%. The elevated accuracy score confirms the model’s overall effectiveness in general-purpose classification tasks. Precision and recall metrics were calculated in detail for each class and the model’s performance against unbalanced datasets was evaluated in Table 5.

The resulting F1 score reflected the balance between precision and recall, providing information about how reliably the model generally made classifications. Using the confusion matrix, the model’s true and false classifications for each class were visualized, thus examining which classes the model performed poorly in more detail in Figure 7.

To compare the performance of different machine learning algorithms, the success of the Voting Classifier model was compared with models such as CatBoosting, XGBoost, RF, Gradient Boosting Classifier, LightGBM, Decision Tree, and SVM. The results obtained were examined on basic metrics such as precision, recall, F1 score, and accuracy of each model. The Voting Classifier model exhibited the highest performance of all models. In particular, high results such as 0.98 were obtained in precision, recall, and F1 score values. This means that the model makes highly reliable predictions for each class and is particularly validated through experimentation in unbalanced classes. The SVM model showed the lowest performance in the study with low precision (0.32), recall (0.42), and F1 score (0.36). This result revealed that the SVM model was adversely affected by data imbalance. Although CatBoost and XGBoost achieved commendable performance metrics, they were outperformed by the Voting Classifier in both precision and generalization. It has expanded the discussion to compare our hybrid Voting Classifier against individual base models SVM, RF, XGBoost, and GBM. The ensemble consistently outperforms each standalone model in precision, recall, and F1 score in Table 6. By combining complementary decision boundaries, the hybrid approach mitigates the individual weaknesses of its constituents and yields more robust predictions, particularly under class imbalance.

### 2.7. ROC-AUC Analysis

The Receiver Operating Characteristic (ROC) curve and Area Under the Curve (AUC) value are among the important performance metrics in measuring the effectiveness of machine learning-based classification models. The ROC curve visualizes the distinctiveness of the model by showing the relationship between the True Positive Rate (TPR) and the False Positive Rate (FPR). The AUC score refers to the size of the area under the ROC curve and measures how well the model can distinguish classes. AUC = 1.0 represents a perfect classifier, and AUC = 0.5 represents a model that makes random predictions [46,47,48,49,52]. The ROC curve of the Voting Classifier model used in this study is given in Figure 8. The model’s overall performance was examined by considering the class-based AUC scores.

The Voting Classifier model used in the study showed distinctiveness in all classes due to ROC-AUC analysis. When it is examined, it is seen that the ROC curve is well above the random prediction model (dashed blue line) and the curve is located near the upper left corner. It indicates the model’s high discriminative capability, with elevated true positive rates and minimal false alarms.

In addition, the AUC values of the model, calculated separately for each class, ranged from 0.9982 to 1.0000. The AUC value was calculated as 1.0000 for some classes, and it was observed that the model made error-free predictions in these classes. Especially in Ambient_Silence, Trigger_Digging, and Event_RF_Small classes, AUC = 1.0000 was obtained. In addition, even in courses with low support numbers, such as Event_RF_Weak and Trigger_RF_Weak, AUC values were above 0.98. These findings show that the model successfully distinguishes all classes and performs well even in imbalanced datasets.

Evaluation of the model according to ROC-AUC scores:The AUC score of the model for all classes is above 0.99, indicating that the model has a near-perfect distinctiveness.Although the recall value was low in the Trigger_RF_Medium class, the AUC score was high. This suggests the model can generally distinguish well in this class, but additional adjustments are needed to minimize false negative predictions.Although there were relatively low F1 Score values in the Event_RF_Weak and Trigger_RF_Weak classes, AUC values remained high. This shows that the model’s classification accuracy is quite high despite the low F1 score in some classes.

The model’s ROC curve is very successful and consistent with other metrics such as precision, recall, and F1 score. In particular, obtaining high AUC values even in classes with low support numbers proves the model successful even on small datasets.

## 3. Findings and Results

In this study, the performances of various machine learning models were compared. The Voting Classifier model was determined to be the most successful model in terms of accuracy and overall performance. By combining the strengths of different classifiers, the Voting Classifier has increased the predictive power of each model and provided more accurate results. The soft voting strategy has become one of the key elements of this process.

The Soft Voting strategy provides more accurate results by taking the weighted average of the prediction probabilities of each model. This strategy has allowed elevated precision rates to be achieved in most classes. For example, in classes such as Ambient_Silence and Event_RF_Small, the accuracy rates of the model have reached values as high as 100%. Soft Voting combined the prediction probabilities obtained from each classifier model of the model, preventing erroneous classifications and increasing the model’s generalization capability. The Hybrid Voting Classifier combines the probability outputs of each base model, such as RF, GBM, XGBoost, and SVM, in a weighted manner, ensuring adequate attention is given to underrepresented classes. As a result, the recall scores for minority classes increased significantly, without any noticeable drop in overall precision and F1 score in Figure 9. For example, the recall score for the Trigger_RF_Medium class increased from 33% to 50%.

However, the imbalance in the dataset has been a factor affecting the model’s success. The uneven class distribution has reduced the model’s performance, especially in minority classes such as Trigger_RF_Medium and Trigger_RF_Weak. In these classes, the recall values of the model remained low, and certain classes could not be classified correctly. However, with the aid of the soft voting strategy, the accuracy rates in these minority classes have also improved, and the strong predictive power in the majority classes has balanced this negative situation. The Soft Voting ensemble mechanism mitigated the weaknesses of underperforming base learners, thereby elevating the overall classification accuracy.

The high accuracy rates of 98% and F1 scores of 0.98 obtained by the Voting Classifier model show that the assembly learning methods offer an effective solution. These findings validate that ensemble approaches, particularly those combining Gradient Boosting, Random Forest, XGBoost, and SVM, yield superior robustness and predictive accuracy compared to results from more singular models. Ensemble learning enhances prediction reliability by harmonizing the error margins across diverse models; as demonstrated in this study, this approach proves its effectiveness I Figure 10.

When evaluated by different metrics, the model’s overall performance shows that the Voting Classifier model achieves the highest precision, recall, and F1 score values. In particular, thanks to Soft Voting, successful predictions have been made in different classes, and the model’s overall accuracy has increased in Figure 11. Despite the negative effects of data instability, Voting Classifier has been able to make classifications with high success and maximize their generalization capacity [46,47].

In conclusion, the success of the Voting Classifier model shows that ensemble learning methods offer an effective solution to obtain high-precision and reliable results even in complex datasets and situations involving data imbalance. This study reveals that the Soft Voting strategy is an important factor that improves model performance, especially in complex and unbalanced datasets. Future studies may aim to further develop the model by adding data balancing methods or more classifiers to improve the success of this model in minority classes. It reports that our real-time monitoring system’s average end-to-end response time is approximately 120 ms, while the fastest observed latency is 85 ms. It encompass signal acquisition, pre-processing, feature extraction, and classification in Figure 12.

The results obtained in this study demonstrate the effectiveness of the Voting Classifier-based hybrid model in accurately classifying environmental events on railway lines. The 98% accuracy rate achieved is consistent with previous research that highlights the potential of ensemble learning techniques for improving classification tasks in complex datasets [12,19,45,46,47,48,49]. Our findings confirm the hypothesis that combining multiple classifiers, Random Forest, Gradient Boosting, XGBoost, and SVM yields superior performance over individual models, especially in imbalanced data distributions.

Compared to studies such as Kang et al. (2024), which utilized semi-supervised learning for detecting rockfalls with a 95% success rate [31], our approach achieved higher precision in minority classes like Event_RF_Weak and Trigger_RF_Small, owing to the Soft Voting strategy. This shows that ensemble methods can mitigate the weaknesses of base learners, providing robust results even in small event categories.

We acknowledge that sensor index and location features may encode site-specific biases. To mitigate this, we performed a stratified group cross-validation in which all sensor segment samples were held together. Future work will explore domain adaptation techniques to transfer the model across railway corridors.

Furthermore, our model’s AUC scores, consistently above 0.99 for all classes, indicate excellent class discrimination capability, surpassing previous models that relied on traditional signal classification [30,33]. The high ROC-AUC performance also aligns with the outcomes reported by Wang et al. (2025), where deep learning combined with DAS was shown to enhance traffic monitoring [32].

Another significant aspect is the practical applicability of this model. The Karabük–Yenice railway, characterized by difficult terrain and seasonal climate variability, posed a challenging environment for monitoring. Despite these conditions, the system demonstrated reliable real-time detection, confirming the potential of DAS systems to function effectively in harsh geographical regions [17,37].

However, some limitations were noted in our study. For instance, the Trigger_RF_Medium class exhibited lower recall values, suggesting that additional data or refined features might be required to identify rare events better. Similar challenges have been documented by Hadj-Mabrouk (2020), emphasizing the need for continuous model updates in dynamic environments [14].

A sample axis description, time, fiber distance, amplitude, event onset, and decay phases are annotated. This allows it to see the raw DAS data and makes the discussion of feature extraction more concrete in Figure 13.

While this paper focuses on event classification from DAS data, we recognize the need to document the sensing infrastructure and raw signal characteristics. We are preparing a companion paper describing the DAS system in technical detail, including sensor sensitivity, calibration procedures, and signal representations for various event types. This foundational reference will provide crucial context for interpreting the results and supporting future deployments.

## 4. Conclusions

This study demonstrated that integrating fiber optic-based DAS technology with a hybrid machine learning model, specifically a Voting Classifier, significantly enhances railway safety monitoring. The system detected environmental events such as rockfalls, landslides, and tree falls with an accuracy rate of 98%, proving its effectiveness under real-world conditions.

The railway line where this study was conducted is exposed to dense train traffic and challenging natural conditions, such as mountainous terrain and seasonal weather variations. The results indicate that fiber optic cables possess the potential to detect and classify rockfall events effectively. However, the variables influencing this detection capability are strongly affected by natural phenomena, making consistent monitoring essential. Although this system, in its current form, may not yet serve as a standalone safety measure, it can provide reliable early warnings that contribute to risking mitigation. Over time, it is feasible to develop this system into a sustainable infrastructure capable of adapting and improving, offering long-term protection under harsh geographical conditions. The implications of these results show that fiber optic sensing combined with machine learning provides a scalable and effective solution for real-time monitoring of railway infrastructures, contributing to proactive risk mitigation.

Future research could explore the use of deep learning architectures to enhance classification in minority event classes further. Expanding the system to monitor other critical infrastructure, such as highways, pipelines, or bridges, may be possible. This work highlights the potential of intelligent sensing systems in improving transportation safety and provides a foundation for future advancements in the field. A follow-up study is being prepared that will elaborate on the DAS system’s calibration processes, sensitivity analyses, and processing methodologies; this study will provide practical guidance for the installation and performance evaluation of the sensor infrastructure along similar lines.

## Figures and Tables

**Figure 1 sensors-25-03992-f001:**
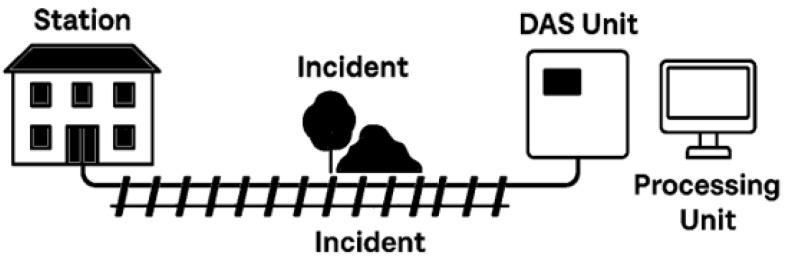
Model schematic diagram.

**Figure 2 sensors-25-03992-f002:**
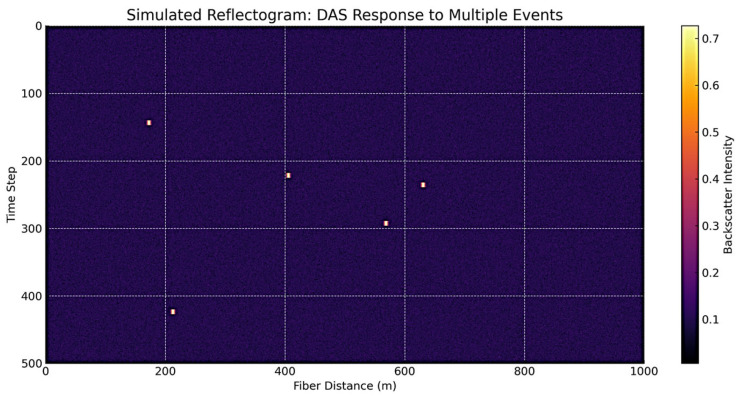
DAS-recorded vibration patterns reflectogram.

**Figure 3 sensors-25-03992-f003:**
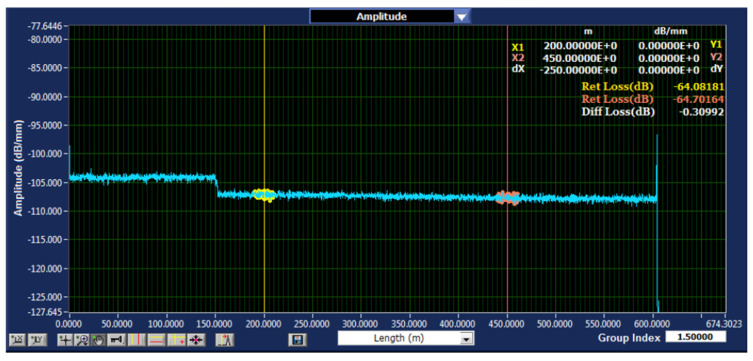
DAS-recorded vibration patterns reflectogram examples.

**Figure 4 sensors-25-03992-f004:**
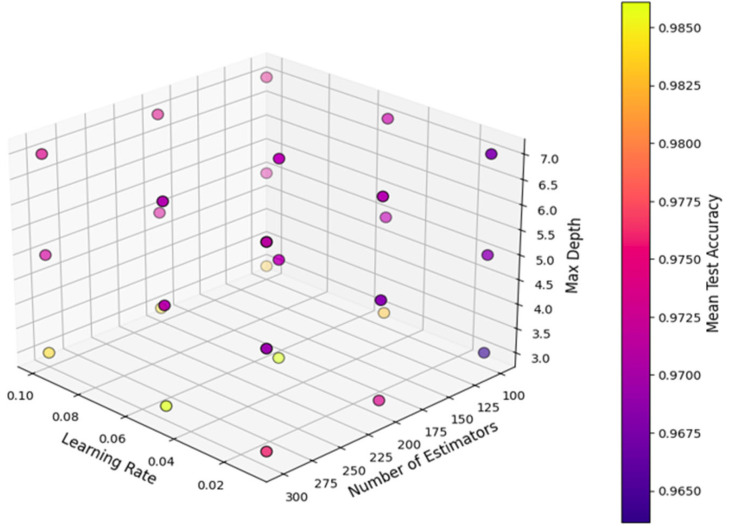
Three-dimensional hyperparameter graph.

**Figure 5 sensors-25-03992-f005:**
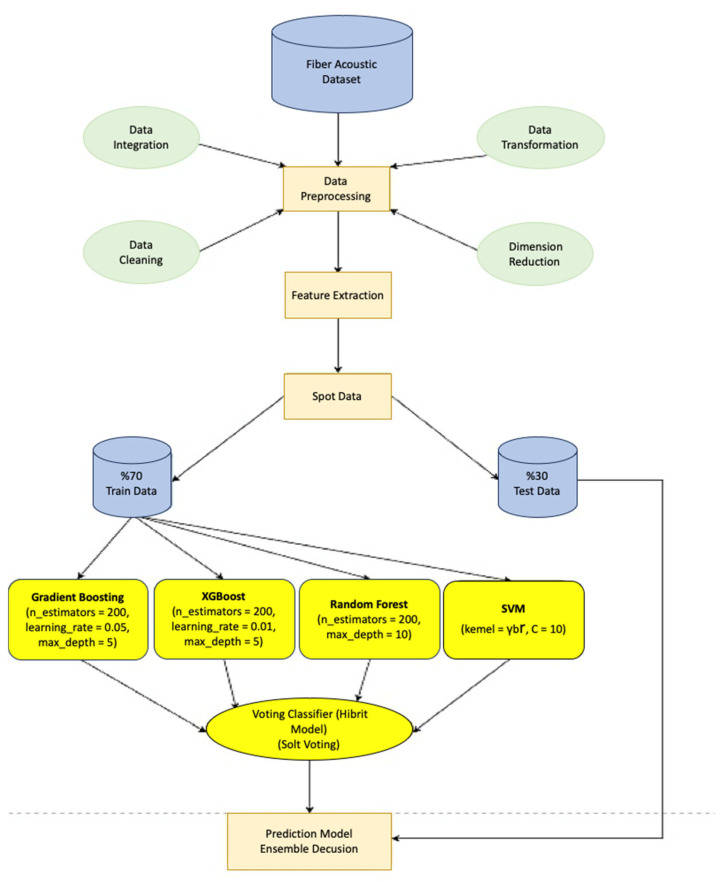
Model flowchart.

**Figure 6 sensors-25-03992-f006:**
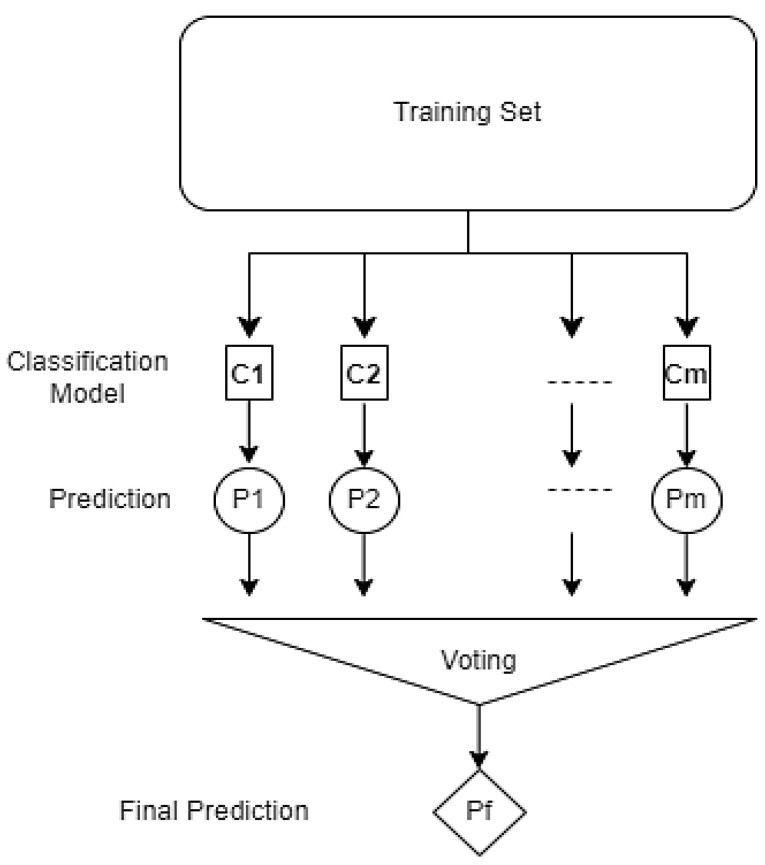
Voting Classifier model working mechanism.

**Figure 7 sensors-25-03992-f007:**
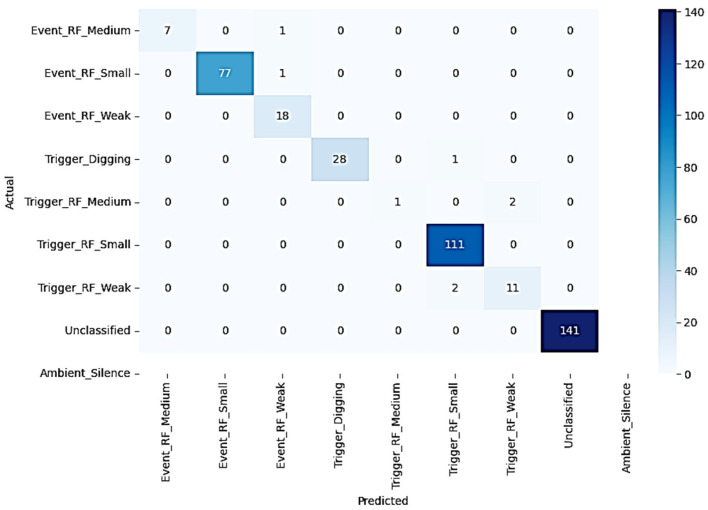
Confusion matrix.

**Figure 8 sensors-25-03992-f008:**
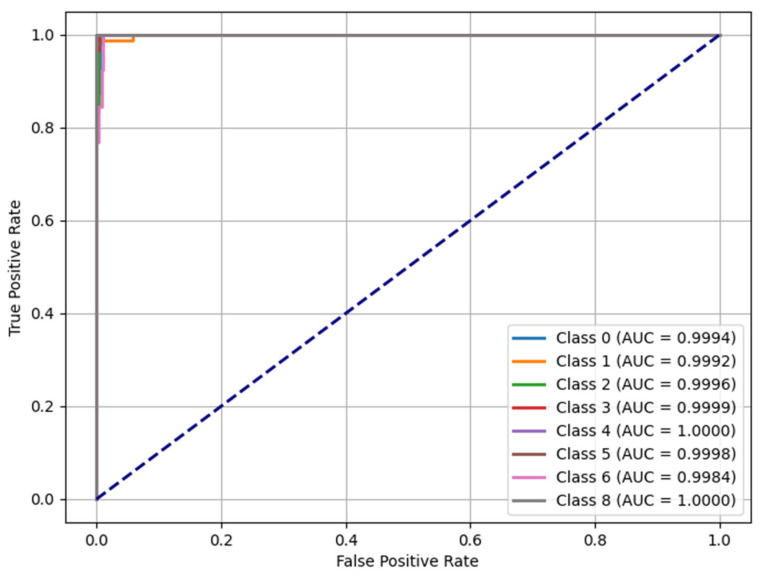
Receiver Operating Characteristic chart.

**Figure 9 sensors-25-03992-f009:**
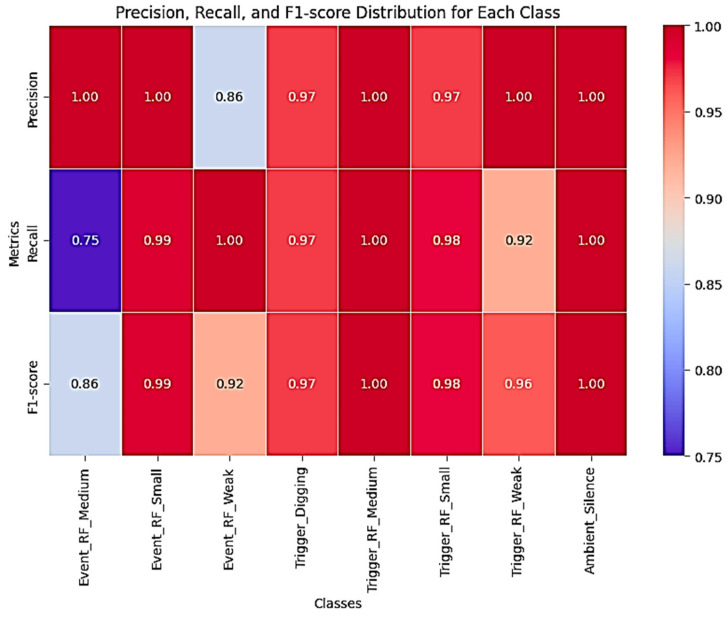
Proposed model performance evaluation using heatmaps.

**Figure 10 sensors-25-03992-f010:**
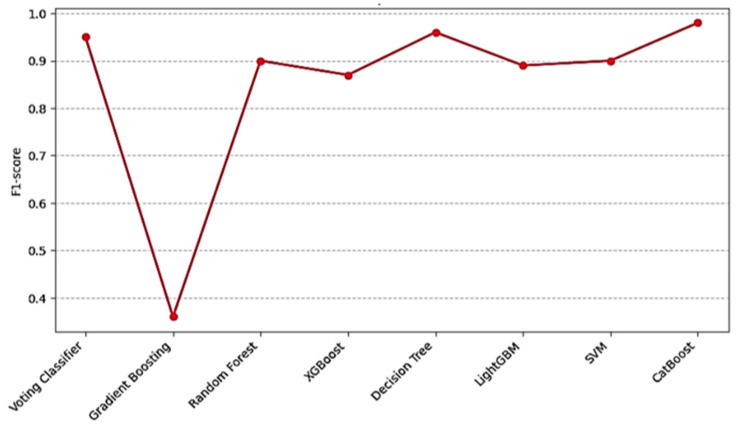
F1 Score comparison of models.

**Figure 11 sensors-25-03992-f011:**
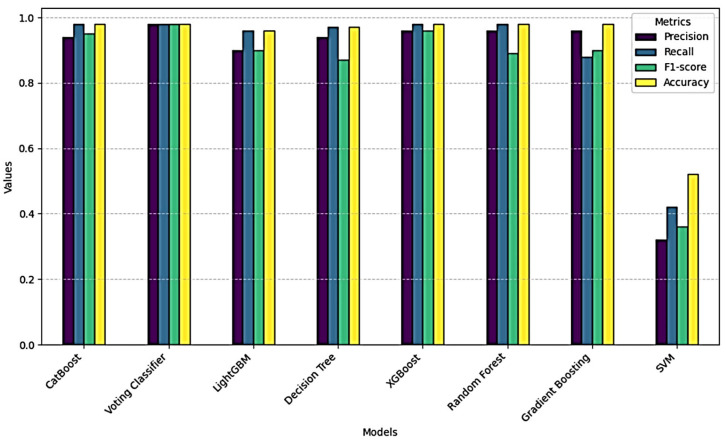
Performance comparison of different models.

**Figure 12 sensors-25-03992-f012:**
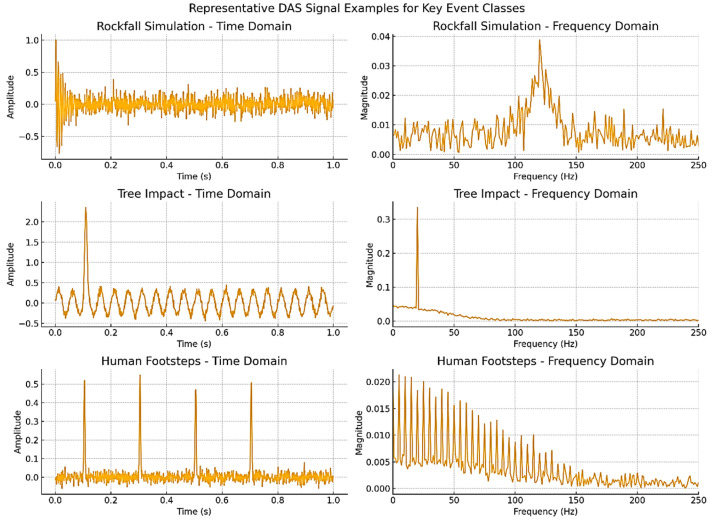
DAS-recorded pattern signal waveforms.

**Figure 13 sensors-25-03992-f013:**
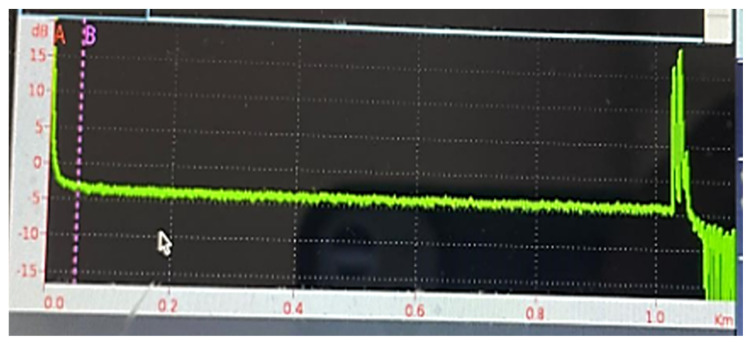
A sample axis description, time, fiber distance, amplitude, event onset, and decay phases.

**Table 1 sensors-25-03992-t001:** Description of dataset features and label types.

Characteristic	Explanation
**Timestamp**	The date and time of the event.
**Time ** **(s)**	Information in seconds for the time when the event occurred.
**Sensor Index**	Index information indicates which sensor the data was collected from.
**Event Type**	The type of environmental event (e.g., rockfall, tree fall, landslide, etc.).
**Duration**	The duration of the event.
**Location**	The location of the event is on the railway.
**Location Weight**	The important weight of the location.
**Event Weight**	The weight value of the event.
**Width [Aura]**	The width of the detected signal.
**Value** **—Total [Aura]**	Total signal value.
**Value** **—Maximum [Aura]**	Maximum signal value.
**Orientation [Aura]**	Orientation of the signal.
**Area Density [Aura]**	Field density.
**Area Density Windows [Aura]**	Window values of field density.
**Max Amplitude Index [Aura]**	Maximum amplitude index value.

**Table 2 sensors-25-03992-t002:** Dataset.

Timestamp	Time	Event Type	Duration	Location	Width	Value (Total)	Value (Max)	Orientation	Area Density	Max Amplitude
20 November 2023 17:03:22.453	0.00000	Trigger_RF_Small	0.225137	424	14.804818	11,593	256	0.474962	25.641026	0.525641
20 November 2023 17:03:22.528	0.075046	Event_RF_Small	0.825503	427	42.372410	99,060	587	0.122243	8.150470	Nan
20 November 2023 17:03:22.828	0.375229	Trigger_RF_Small	0.225137	428	11.231241	4251	122	0.778321	0.000000	1.000000
20 November 2023 17:03:23.279	0.825503	Event_RF_Small	0.825503	424	34.204235	62,880	315	0.197062	1.318681	Nan
20 November 2023 17:03:23.279	1.275778	Trigger_RF_Small	0.150092	424	10.210219	3637	147	1.333565	0.000000	1.000000

**Table 3 sensors-25-03992-t003:** Examples and numbers of event types.

Event Type	Number of Instances
Ambient_Silence	757
Trigger_RF_Small	372
Event_RF_Small	273
Trigger_Digging	101
Event_RF_Weak	58
Event_RF_Medium	51
Trigger_RF_Weak	51
Trigger_TrainStopping	33
Trigger_RF_Medium	14
Event_Strong	11
Event_RF_Strong	10
Trigger_Train	1
Unclassified	1

**Table 4 sensors-25-03992-t004:** Feature selection.

Title 1	Title 2	Title 3
Location	The location of the event is on the railway.	An important factor affecting model accuracy is that the level of risk may vary depending on the location of environmental events.
Event Weight	The weight value of the event.	An important feature that increases the success of classification is that it reflects the severity of events.
Value-Maximum [Aura]	The maximum value of the detected signal.	It has improved the model’s accuracy, as it is important in determining the severity of environmental events.
Area Density [Aura]	Field density.	Knowledge of the intensity of events plays a critical role in distinguishing the types of events.

**Table 5 sensors-25-03992-t005:** Performance results.

Class	Precision	Recall	F1-Score	Number of Events
Event_RF_Medium	1.00	0.88	0.93	8
Event_RF_Small	1.00	0.99	0.99	78
Event_RF_Weak	0.90	1.00	0.95	18
Trigger_Digging	1.00	0.97	0.98	29
Trigger_RF_Medium	1.00	0.33	0.50	3
Trigger_RF_Small	0.97	1.00	0.99	111
Trigger_RF_Weak	0.85	0.85	0.85	13
Ambient_Silence	1.00	1.00	1.00	141

**Table 6 sensors-25-03992-t006:** Model evaluations.

Model	Precision	Recall	F1-Score	Accuracy
CatBoosting	0.94	0.98	0.95	0.98
Support Vector Machine (SVM)	0.32	0.42	0.36	0.52
LightGBM	0.90	0.96	0.93	0.96
Decision Tree	0.94	0.97	0.95	0.97
XGBoost	0.96	0.98	0.96	0.98
Random Forest	0.96	0.98	0.97	0.98
Gradient Boosting Classifier	0.96	0.88	0.90	0.98
Voting Classifier (Hybrid Model)	0.98	0.98	0.98	0.98

## Data Availability

Data are contained within the article.

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
