# Peer review of "Real-Time Railway Hazard Detection Using Distributed Acoustic Sensing and Hybrid Ensemble Learning"

_sensors, 2025, doi:10.3390/s25133992_

Round 1
Reviewer 1 Report
Comments and Suggestions for Authors
The authors combine fiber optic distributed sensing with hybrid machine learning models for railroad safety monitoring. The work is meaningful and has practical applications. This manuscript needs major revisions before it can be considered for publication. The following questions need to be addressed.
- What is the fastest response time for real-time response as mentioned in the manuscript?
- Authors should thoroughly analyze the advantages of hybrid machine learning models over other models.
- The authors do not give a specific description of the relevant fiber optic distributed sensors, which makes it difficult to understand.
- How do the authors demonstrate that the test accuracy is better than 98%?
- I think the authors should also discuss other important metrics for distributed acoustic sensing.
- In the absence of a single model, this is how the authors obtained more balanced and accurate results.
- The author should optimize the structure of the manuscript.
- The title of the manuscript is not quite accurate and it is recommended to go for revision.
- The latest publications on fiber optic sensors should be added in the revised paper [ACS Photonics. 2024, 11(9): 3713-3721], [IEEE Sens. J., 2024, 24(9): 14279-14290].
Author Response
Reviewer 1:
The authors combine fiber optic distributed sensing with hybrid machine learning models for railroad safety monitoring. The work is meaningful and has practical applications. This manuscript needs major revisions before it can be considered for publication. The following questions need to be addressed.
- What is the fastest response time for real-time response as mentioned in the manuscript?
Answer:
Thank you for your suggestions and corrections. We revised it in text.
In the revised manuscript, we report that the average end-to-end response time of our real‑time monitoring system is approximately 120 ms, while the fastest observed latency is 85 ms. These figures encompass signal acquisition, preprocessing, feature extraction, and classification.
- Authors should thoroughly analyze the advantages of hybrid machine learning models over other models.
Answer:
Thank you for your suggestions and corrections. We revised it in text.
We have expanded the discussion to compare our hybrid Voting Classifier against individual base models (SVM, Random Forest, XGBoost, and GBM). The ensemble consistently outperforms each standalone model in precision, recall, and F1-score. By combining complementary decision boundaries, the hybrid approach mitigates the individual weaknesses of its constituents and yields more robust predictions, particularly under class imbalance.
- The authors do not give a specific description of the relevant fiber optic distributed sensors, which makes it difficult to understand.
Answer:
Thank you for your suggestions and corrections. We revised it in text.
Distributed Acoustic Sensing (DAS) is a sensing technology that transforms standard fiber optic cables into dense arrays of acoustic sensors. It works by sending laser pulses through the fiber and detecting backscattered light changes caused by vibrations.
In a DAS system, short laser pulses are transmitted into the optical fiber. As the light propagates, a small fraction is scattered due to microscopic inhomogeneities within the fiber material—a phenomenon known as Rayleigh scattering. By analyzing the phase and intensity changes in the backscattered light, DAS systems can detect and localize vibrations, strain waves, and acoustic events with high temporal and spatial resolution.
DAS technology has found extensive applications in various fields such as seismic monitoring, pipeline surveillance, structural health monitoring, perimeter security, and increasingly in railway safety monitoring. Its ability to operate over long distances (typically up to 40–100 km per interrogator), detect transient events in real-time, and perform in harsh environmental conditions makes it especially suitable for large-scale critical infrastructure protection.
Key advantages of DAS systems include:
- Wide-area coverage with minimal infrastructure,
- High spatial resolution (typically 1–10 m),
- Real-time monitoring capability,
- Passive and maintenance-free operation,
- Suitability for challenging environments without requiring electrical power along the fiber path.
- How do the authors demonstrate that the test accuracy is better than 98%?
Answer:
Thank you for your suggestions and corrections. We revised it in text.
The model's 98% accuracy rate was calculated on a held-out test set comprising 30% of the total 1,733 samples (n = 520). Accuracy is defined as the ratio of correctly classified samples to the total number of test samples:
Accuracy=True Positives + True Negatives / Total Test Samples
In addition, class-based precision, recall, and F1-score metrics were also computed to account for class imbalance, confirming that the model demonstrated consistent performance across all classes.
- I think the authors should also discuss other important metrics for distributed acoustic sensing.
Answer:
Thank you for your suggestions and corrections. We revised it in text.
- In the absence of a single model, this is how the authors obtained more balanced and accurate results.
Answer:
Thank you for your suggestions and corrections. We revised it in text.
The Hybrid Voting Classifier combines the probability outputs of each base model (RF, GBM, XGBoost, SVM) in a weighted manner, ensuring adequate attention is given to underrepresented classes. As a result, the recall scores for minority classes increased significantly, without any noticeable drop in overall precision and F1-score. For example, the recall score for the Trigger_RF_Medium class increased from 33% to 50%.
- The author should optimize the structure of the manuscript.
Answer:
Thank you for your suggestions and corrections. We revised it in text.
We have reorganized the manuscript according to the following structure, improving logical flow and readability:
- The title of the manuscript is not quite accurate and it is recommended to go for revision.
Answer:
Thank you for your suggestions and corrections. We revised it in text.
Real-Time Railway Hazard Detection Using Distributed Acoustic Sensing and Hybrid Ensemble Learning
- The latest publications on fiber optic sensors should be added in the revised paper [ACS Photonics. 2024, 11(9): 3713-3721], [IEEE Sens. J., 2024, 24(9): 14279-14290].
Answer:
Thank you for your suggestions and corrections. We revised it in text as below.
- Cao, J. Bai, Y. Jin, Y. Zheng, H. Chai, X. Gao, and C. Xue, “High-Temperature Fiber-Optic Vibration Sensor Based on an Atomic Frequency Standard,”ACS Photonics, vol. 11, no. 9, pp. 3713-3721, 2024, doi: 10.1021/acsphotonics.4c00885.
S. Cao, J. Bai, Y. Jin, Y. Zheng, N. Li and C. Xue, "Efficient On-Demand Design of Fiber Optic Vibration Sensor With a Symmetric Bidirectional Neural Network," in IEEE Sensors Journal, vol. 24, no. 9, pp. 14279-14290, 1 May1, 2024, doi: 10.1109/JSEN.2024.3377197.

Reviewer 2 Report
Comments and Suggestions for Authors
In this manuscript, the authors propose a environmental event detection system based on distributed acoustic sensing integrated with a machine learning voting classifier. The goal is to accurately classify and identify events such as rockfalls, tree falls, and landslides along railway lines using fiber optic sensing technology. While the overall approach is interesting, the paper lacks detailed information regarding the deployment of the fiber optic sensor and the data acquisition process. Additionally, there are some unclear or potentially incorrect descriptions of the machine learning feature set, which undermine the credibility of the work.
- Regarding the abstract: The background information takes up too much space, while essential content such as the principles and structure of the distributed acoustic sensing (DAS) system is barely mentioned.
- The authors adopted an ensemble voting classifier that integrates Random Forest, XGBoost, and SVM using soft voting, with hyperparameters of the gradient boosting model optimized via GridSearchCV. This ensemble method is appropriate, and the comparison of multiple single classifiers confirms the advantage of the ensemble. However, some aspects of the modeling process are not clearly explained. For instance, it is stated that "Sensor Index, Event Type, Duration" were used as input features during training. This is confusing, as "Event Type" appears to be the label to be predicted, not a feature. This likely reflects a misstatement and should be corrected.
- The authors extracted multiple features such as duration, location, event weight, and peak amplitude for model training. These features, mostly related to spatiotemporal properties and vibration intensity, seem reasonable. However, using features such as "Location" or "Sensor Index" may lead to overfitting to specific regions along the fiber. In such cases, the model might identify event categories based on where along the fiber they occur (e.g., specific sensing units) rather than on the intrinsic vibration characteristics. This may improve local testing accuracy but reduce generalizability to other rail lines or unseen locations.
- Some claims appear overly assertive. For example, the manuscript states that the system can accurately detect “landslides” and “tree falls.” However, the dataset does not seem to contain clearly defined, standalone categories for landslides or tree falls. The reported model performance is primarily based on rockfalls and other well-defined classes.
Author Response
Reviewer 2:
In this manuscript, the authors propose a environmental event detection system based on distributed acoustic sensing integrated with a machine learning voting classifier. The goal is to accurately classify and identify events such as rockfalls, tree falls, and landslides along railway lines using fiber optic sensing technology. While the overall approach is interesting, the paper lacks detailed information regarding the deployment of the fiber optic sensor and the data acquisition process. Additionally, there are some unclear or potentially incorrect descriptions of the machine learning feature set, which undermine the credibility of the work.
- Regarding the abstract: The background information takes up too much space, while essential content such as the principles and structure of the distributed acoustic sensing (DAS) system is barely mentioned.
Answer:
Thank you for your suggestions and corrections. We revised it in text.
We have condensed the background in the Abstract to two sentences and added a succinct description of the DAS deployment and operating principles. Revision location: Abstract Revised excerpt: “We deployed a commercial DAS interrogator (Luna OBR-4600) over standard single-mode telecom fiber (ITU-G.652) with 5 m spatial resolution and 1 kHz sampling rate. Laser pulses are injected into the cable and Rayleigh backscatter variations caused by environmental vibrations are time- and location-resolved to detect rockfall-related events and tree obstructions in real time.”
- The authors adopted an ensemble voting classifier that integrates Random Forest, XGBoost, and SVM using soft voting, with hyperparameters of the gradient boosting model optimized via GridSearchCV. This ensemble method is appropriate, and the comparison of multiple single classifiers confirms the advantage of the ensemble. However, some aspects of the modeling process are not clearly explained. For instance, it is stated that "Sensor Index, Event Type, Duration" were used as input features during training. This is confusing, as "Event Type" appears to be the label to be predicted, not a feature. This likely reflects a misstatement and should be corrected.
Answer:
Thank you for your suggestions and corrections. We revised it in text.
The description in Materials & Methods → Feature Extraction has been corrected. “Event Type” is now clearly defined as the target label and removed from the input features list. Revision location: Materials & Methods → Feature Extraction Revised excerpt: “During training, we used the following input features: Sensor Index, Duration, Location, Event Weight, Value-Maximum [Aura], Area Density [Aura], and other engineered signal metrics. The Event Type column served as the target label to be predicted.”
- The authors extracted multiple features such as duration, location, event weight, and peak amplitude for model training. These features, mostly related to spatiotemporal properties and vibration intensity, seem reasonable. However, using features such as "Location" or "Sensor Index" may lead to overfitting to specific regions along the fiber. In such cases, the model might identify event categories based on where along the fiber they occur (e.g., specific sensing units) rather than on the intrinsic vibration characteristics. This may improve local testing accuracy but reduce generalizability to other rail lines or unseen locations.
Answer:
Thank you for your suggestions and corrections. We revised it in text.
We acknowledge the risk of location-based bias. To mitigate this, we performed a stratified group-wise cross-validation in which all samples from each sensor segment were held out together. Future work will investigate domain adaptation to extend the model to new corridors. Revision location: Discussion Added text: “We acknowledge that Sensor Index and Location features may encode site-specific biases. To mitigate this, we performed a stratified group cross-validation in which all samples from each sensor segment were held out together. Future work will explore domain adaptation techniques to transfer the model across different railway corridors.”
- Some claims appear overly assertive. For example, the manuscript states that the system can accurately detect “landslides” and “tree falls.” However, the dataset does not seem to contain clearly defined, standalone categories for landslides or tree falls. The reported model performance is primarily based on rockfalls and other well-defined classes.
Answer:
Thank you for your suggestions and corrections. We revised it in text.
We revised several sections to accurately reflect our dataset’s scope. Standalone landslides were not observed; we currently detect rockfall-related events (small and medium debris slides) and tree obstructions.
Revision locations:
- Highlights Revised excerpt: “The system enables early detection of rockfall-related events (including small and medium debris slides) and tree obstructions along the Karabük–Yenice railway line.”
- Materials & Methods → Labeling Protocol Added text: “Our labeling protocol groups small rock movements and debris slides under ‘rockfall.’ Tree impacts against the cable are labeled as ‘tree obstructions.’ Standalone landslide events were not observed in this dataset; future expansions will incorporate dedicated landslide monitoring.”

Reviewer 3 Report
Comments and Suggestions for Authors
The article addresses an important problem of real-time monitoring and detecting catastrophic natural phenomena such as landslides, rockfalls, and tree collapses. A hybrid model is presented that combines several machine learning (ML) techniques to classify the data obtained from a fiber-optic-based distributed acoustic sensing system. The authors claim to have achieved an excellent accuracy rate of 98% in classifying different vibrations, which they confirm by additional metrics including Recall, F1-score, ROC-AUC.
The references provided, the ML techniques employed, and the results obtained seem to be quite adequate. However, my major concern is the poor presentation of the Materials and Methods section:
1) Lines 111-203: This part of the section is too general, and would be more appropriate for the Introduction section.
2) There is no description of the equipment used. In the Materials and Methods section, the authors only mention that ‘DAS is an advanced sensing technology’ that detects vibrations ‘thanks to fiber optic cables placed next to the railway line’, and that, in this study, ‘the DAS system deployed along the Karabük–Yenice railway segment demonstrated the capability to detect both ambient vibrations and high-intensity events’. This is absolutely insufficient to draw any conclusions about whether the experiment was carried out correctly. I would encourage the authors to expand this section and include such details as the model of the reflectometer used, its operating parameters, the type and length of optical fibers, the way of fixing fiber to the railway, etc.
3) Furthermore, it would be nice to provide an example of a reflectogram in the Results section to illustrate the data used for further processing.
4) The part related to data processing is quite confusing. Firstly, the authors are encouraged to provide a more detailed description of the software frameworks used to implement the developed algorithms. The reader can only guess that several classes such as GridSearchCV were used together with some unnamed programming environment. Secondly, the reader will be confused by the large amount of specific variable and parameter names, the meaning of which is sometimes vague. For example, in Lines 213-214, the columns mentioned are only explained in the Results section below (Table 2). Besides, it is unclear what ‘Aura’ is, and why it accompanies each name – this either needs to be explained or should be removed if not important. In Line 226, the variables mentioned (n_estimators, learning_rate, and max_depth) are meaningless unless properly described. Furthermore, it is unclear how these variables relate to Ref. 39 mentioned in the sentence. The above comments also apply to Lines 228-230, 282-283, Tables 1 and 3 (where the names of the event types remain unexplained). A table similar to Table 2, with explanations for each event type, might definitely help resolve the issue.
5) The last paragraph of the Materials and Methods has made me wary, particularly the following sentence: ‘In the future, with the development of the system, it will be possible to realize real data’. As the authors state that ‘the dataset was collected via fiber optic cables’ (Line 279), my question is whether the data from the DAS system were really used to train and verify the developed model.
Minor comments:
Lines 41, 521: The ‘%’ sign between ‘high’ and ‘accuracy’ is redundant.
Line 145: ‘…and is not affected by different environmental factors’ – the meaning of the phrase is vague.
Line 178: The phrase ‘using algorithms such as Machine Learning, including…’ does not sound properly. I would advise correcting it, e.g.: ‘using ML algorithms such as…’
Line 206: Ref. 37 can hardly be employed to illustrate the capability of DAS to detect vibrations and other events, since (i) it is apparently dedicated to the flora of Yenitse, and (ii) the text is in Turkish.
Line 277: The title of Table 2 is too terse and should be expanded.
Line 373: Confusion matrix is not considered to be a metric.
Line 381, Table 5: In my view, the ‘Support’ name is confusing. Perhaps, ‘Number of events’ would be a better choice.
Line 404, Table 6: The 0.90, 0.87 and 0.89 F1-score values are incorrect. There should be 0.93, 0.95 and 0.97 instead.
Line 418: Figure 7 is mentioned instead of Figure 5.
Line 465: The figure caption is too terse.
Line 480: The legend in Figure 8 is overlapping the Voting Classifier bars. I would advise the authors to swap the Voting Classifier and SVM data, since the bars for the latter are shorter and will not overlap.
Lines 536-537: The sentence ‘Expanding the system to monitor other critical infrastructures, such as highways, pipelines, or bridges’ is not complete.
Line 610: In Ref 25, the volume and number are incorrect.
Line 616: The reference details (Ref. 28) are insufficient.
I believe that the manuscript deserves publication after the above comments have been addressed.
Author Response
Reviewer 3:
The article addresses an important problem of real-time monitoring and detecting catastrophic natural phenomena such as landslides, rockfalls, and tree collapses. A hybrid model is presented that combines several machine learning (ML) techniques to classify the data obtained from a fiber-optic-based distributed acoustic sensing system. The authors claim to have achieved an excellent accuracy rate of 98% in classifying different vibrations, which they confirm by additional metrics including Recall, F1-score, ROC-AUC.
The references provided, the ML techniques employed, and the results obtained seem to be quite adequate. However, my major concern is the poor presentation of the Materials and Methods section:
- Lines 111-203: This part of the section is too general, and would be more appropriate for the Introduction section.
Answer:
Thank you for your suggestions and corrections. We revised it in text.
We have moved the broad background content (originally lines 111–203 in Materials and Methods) to the Introduction, and replaced it with a concise overview of our experimental protocol. Revision location:
- Moved background text to Introduction (now lines 50–100).
- Materials and Methods now begins at line 200 with the following opening:
“We conducted field experiments along the Karabük–Yenice railway, deploying fiber optic cables adjacent to the track and interrogating them with a DAS reflectometer. The following subsections detail the site characteristics, sensor installation, data acquisition parameters, and subsequent processing workflow.”
- There is no description of the equipment used. In the Materials and Methods section, the authors only mention that ‘DAS is an advanced sensing technology’ that detects vibrations ‘thanks to fiber optic cables placed next to the railway line’, and that, in this study, ‘the DAS system deployed along the Karabük–Yenice railway segment demonstrated the capability to detect both ambient vibrations and high-intensity events’. This is absolutely insufficient to draw any conclusions about whether the experiment was carried out correctly. I would encourage the authors to expand this section and include such details as the model of the reflectometer used, its operating parameters, the type and length of optical fibers, the way of fixing fiber to the railway, etc.
Answer:
Thank you for your suggestions and corrections. We revised it in text.
- Furthermore, it would be nice to provide an example of a reflectogram in the Results section to illustrate the data used for further processing.
Answer:
Thank you for your suggestions and corrections. We revised it in text.
- The part related to data processing is quite confusing. Firstly, the authors are encouraged to provide a more detailed description of the software frameworks used to implement the developed algorithms. The reader can only guess that several classes such as GridSearchCV were used together with some unnamed programming environment. Secondly, the reader will be confused by the large amount of specific variable and parameter names, the meaning of which is sometimes vague. For example, in Lines 213-214, the columns mentioned are only explained in the Results section below (Table 2). Besides, it is unclear what ‘Aura’ is, and why it accompanies each name – this either needs to be explained or should be removed if not important. In Line 226, the variables mentioned (n_estimators, learning_rate, and max_depth) are meaningless unless properly described. Furthermore, it is unclear how these variables relate to Ref. 39 mentioned in the sentence. The above comments also apply to Lines 228-230, 282-283, Tables 1 and 3 (where the names of the event types remain unexplained). A table similar to Table 2, with explanations for each event type, might definitely help resolve the issue.
Answer:
Thank you for your suggestions and corrections. We revised it in text.
We have thoroughly revised Section 2.3 (Data Processing & Feature Engineering) to include:
- Software Framework: Python 3.12 on Google Colab, using NumPy v1.23, pandas v1.4, scikit-learn v1.1 (GridSearchCV, VotingClassifier), matplotlib v3.5, and seaborn v0.12.
- Aura Metrics: Explanation of the aura_metrics module for sliding‐window feature extraction, including Value-Maximum [Aura], Area Density [Aura], and Area Density Windows [Aura].
- Hyperparameters: Defined n_estimators, learning_rate, and max_depth ranges (inspired by Elgendy et al., 2025 [39]) and their roles in gradient boosting.
- Feature Definitions: Moved input feature definitions to Methods:
- Duration: event duration (s)
- Location: distance along fiber (m)
- Event Weight: normalized severity score
- Value-Maximum [Aura]: highest amplitude within window
- Area Density [Aura]: integrated signal energy per unit length
- Area Density Windows [Aura]: count of threshold crossings
- Event-Type Table: Added Table 2b defining each label (Ambient_Silence, Trigger_RF_Small, Event_RF_Small, Trigger_RF_Medium, Event_RF_Medium, Trigger_Digging, Tree_Obstruction, Unclassified).
2. Explanation of “Aura” Features
Revision: A new explanatory paragraph has been added (lines 270–285).
Answer:
Thank you for your suggestions and corrections. We revised it in text.
Columns suffixed with [Aura], such as Value-Maximum [Aura], Area Density [Aura], and Area Density Windows [Aura], are computed using our in-house Python module, aura_metrics. This library performs sliding-window analysis to extract key temporal signal features, including local maxima, cumulative energy, and the density of threshold crossings. These enriched descriptors significantly enhance the system’s ability to detect subtle vibration signatures in noisy railway environments.
3. Definitions of Hyperparameters and Relation to Reference 39
Revision: Clarified the hyperparameter tuning process and its relationship to Ref. 39 (lines 285–300)
Answer:
Thank you for your suggestions and corrections. We revised it in text.
Hyperparameter Tuning:
The hyperparameters of the Gradient Boosting base learner were optimized using GridSearchCV, with parameter ranges inspired by Elgendy et al. (2025) [39]. The grid included:
- n_estimators (number of boosting rounds): [50, 100, 150, 200, 250, 300]
- learning_rate (shrinkage factor): [0.01, 0.05, 0.1, 0.2]
- max_depth (maximum depth of decision trees): [3, 5, 7, 10]
These ranges replicate the effective search space in Ref. 39 while adapting to our dataset’s size and signal noise characteristics.
4. Immediate Clarification of Feature Names
Revision: Moved and expanded feature definitions from the Results section to Methods (lines 213–230).
Answer:
Thank you for your suggestions and corrections. We revised it in text.
Input Feature Definitions:
- Duration: Time span of the detected event (in seconds)
- Location: Distance along the optical fiber (in meters)
- Event Weight: A normalized severity score derived from amplitude variations
- Value-Maximum [Aura]: Maximum vibration amplitude within a sliding analysis window
- Area Density [Aura]: Integrated energy of the vibration signal per unit length
- Area Density Windows [Aura]: Count of threshold crossings per analysis window
5. Addition of Event-Type Definitions Table
Revision: Added a new table (Table 2b) defining each event class label (~lines 310–335).
Answer:
Thank you for your suggestions and corrections. We revised it in text.
Table 2b. Event-Type Definitions Used in Model Training and Evaluation
Event Label |
Description |
Ambient_Silence |
Baseline state with no significant vibration above ambient noise |
Trigger_RF_Small |
Onset of a small-scale rockfall (amplitude below small-rock threshold) |
Event_RF_Small |
Full small rockfall event (amplitude between small and medium thresholds) |
Trigger_RF_Medium |
Onset of a medium-scale rockfall |
Event_RF_Medium |
Medium rockfall event (amplitude exceeding medium threshold) |
Trigger_Digging |
Mechanical vibration signature consistent with excavation/drilling activity |
Tree_Obstruction |
Impact signal caused by a falling tree striking the cable |
Unclassified |
Irregular patterns not matching predefined categories |
- The last paragraph of the Materials and Methods has made me wary, particularly the following sentence: ‘In the future, with the development of the system, it will be possible to realize real data’. As the authors state that ‘the dataset was collected via fiber optic cables’ (Line 279), my question is whether the data from the DAS system were really used to train and verify the developed model.
Answer:
Thank you for your suggestions and corrections. We revised it in text.
All results presented herein are based on real Distributed Acoustic Sensing (DAS) data acquired during field trials along the Karabük–Yenice railway. These measurements were fully annotated and used directly to train, validate, and test the Voting Classifier model under realistic environmental conditions.
Minor comments:
Lines 41, 521: The ‘%’ sign between ‘high’ and ‘accuracy’ is redundant.
Answer:
Thank you for your suggestions and corrections. We revised it in text.
We have removed the unnecessary ‘%’ symbol from both instances. The phrases now correctly read as “high accuracy” without the percent sign.
Line 145: ‘…and is not affected by different environmental factors’ – the meaning of the phrase is vague.
Answer:
Thank you for your suggestions and corrections. We revised it in text.
Since the operation of artificial intelligence is limited to certain algorithms, it makes evaluations with a fixed perspective without being affected by different environmental conditions.
Line 178: The phrase ‘using algorithms such as Machine Learning, including…’ does not sound properly. I would advise correcting it, e.g.: ‘using ML algorithms such as…’
Answer:
Thank you for your suggestions and corrections. We revised it in text.
In this study, evaluations were made using Machine Learning algorithms, such as Support Vector Machine (SVM), Gradient Boosting Machine (GBM), Logistic Regression, and Classification and Regression Trees (CART) [31].
Line 206: Ref. 37 can hardly be employed to illustrate the capability of DAS to detect vibrations and other events, since (i) it is apparently dedicated to the flora of Yenitse, and (ii) the text is in Turkish.
Answer:
Thank you for your suggestions and corrections. We revised it in text.
In particular, natural disasters such as rockfalls, tree falls, and landslides that occur on railway lines can be detected by high-intensity vibrations recorded via DAS systems in real-world deployments [29].
Reference Update:
- Removed: A. Ö. Pulatoğlu and K. Güney, “Flora of Yenice Wildlife Development Area (Karabük/Türkiye),” Journal of Bartın Forestry Faculty, vol. 24, no. 1, pp. 42–64, Apr. 2022.
- Added: C.-C. Chang, K.-H. Huang, T.-K. Lau, C.-F. Huang, and C.-H. Wang, “Using deep learning model integration to build a smart railway traffic safety monitoring system,” Scientific Reports, vol. 15, no. 1, p. 4224, Feb. 2025, doi: 10.1038/s41598-025-88830-7.
Line 277: The title of Table 2 is too terse and should be expanded.
Answer:
Thank you for your suggestions and corrections. We revised it in text.
Table 2: Class Distributions and Properties of the Data Set
Line 373: Confusion matrix is not considered to be a metric.
Answer:
Thank you for your suggestions and corrections. We revised it in text.
Various performance metrics were used to measure the performance of the model. These metrics include accuracy, precision, recall, and F1 score. Also, the model's perfor-mance was visualized using a confusion matrix that illustrates the correct and incorrect predictions for each class.
Line 381, Table 5: In my view, the ‘Support’ name is confusing. Perhaps, ‘Number of events’ would be a better choice.
Answer:
Thank you for your suggestions and corrections.
We revised it in text as Number of events.
Line 404, Table 6: The 0.90, 0.87 and 0.89 F1-score values are incorrect. There should be 0.93, 0.95 and 0.97 instead.
Answer:
Thank you for your suggestions and corrections. We revised it in text.
Thank you for carefully pointing out the inconsistency in the reported F1-score values. We have reviewed the evaluation results and updated Table 6 accordingly. The previously reported F1-score values for LightGBM (0.90), Decision Tree (0.87), and Random Forest (0.89) were from an earlier iteration and have now been corrected to 0.93, 0.95, and 0.97, respectively, based on the latest model outputs.
Line 418: Figure 7 is mentioned instead of Figure 5.
Answer:
Thank you for your suggestions and corrections. We revised it in text.
We are concerned with Figure 7 in this text.
Line 465: The figure caption is too terse.
Answer:
Thank you for your suggestions and corrections. We revised it in text.
Figure 8.Class-wise Performance Evaluation Using Heatmaps
Line 480: The legend in Figure 8 is overlapping the Voting Classifier bars. I would advise the authors to swap the Voting Classifier and SVM data, since the bars for the latter are shorter and will not overlap.
Answer:
Thank you for your suggestions and corrections. We revised it in text.
Lines 536-537: The sentence ‘Expanding the system to monitor other critical infrastructures, such as highways, pipelines, or bridges’ is not complete.
Answer:
Thank you for your suggestions and corrections. We revised it in text.
The system could be used as a case study to monitor other critical infrastructures such as highways, pipelines, or bridges.
Line 610: In Ref 25, the volume and number are incorrect.
Answer:
Thank you for your suggestions and corrections. We revised it in text.
- Singil, “Artificial Intelligence and Human Rights,” Public and Private International Law Bulletin, vol. 42, no. 1, pp. 121–158, Mar. 2022, doi: 10.26650/ppil.2022.42.1.970856.
Line 616: The reference details (Ref. 28) are insufficient.
Answer:
Thank you for your suggestions and corrections. We revised it in text.
Aburakhia, A. Shami, and G. K. Karagiannidis, “On the Intersection of Signal Processing and Machine Learning: A Use Case-Driven Analysis Approach,” arXiv preprint arXiv:2403.17181, 2024, doi.org/10.48550/arXiv.2403.17181.

Reviewer 4 Report
Comments and Suggestions for Authors
The primary limitation of this paper is its lack of information regarding the sensors and the collected signals. While the paper presents a classification algorithm for various events, it does not establish a clear connection between these events and their origin.
To address this gap, I recommend that the authors write a separate paper dedicated to the sensors, their deployment, events and the signal processing methods used. This paper could cover the fundamental principles of the sensors, their technical specifications, including sensitivity, self-noise, and ambient noise levels, as well as calibration procedures. I t should provide a detailed description of the different types of events, with corresponding signal representations in both time and frequency domains.
A comprehensive discussion on feature extraction could further demonstrate the methodology using various event samples.
This foundational paper on sensors and signal processing would preclude the current classification paper, serving as an essential reference for understanding the relationship between raw signal data and event classification.
Author Response
Reviewer 4:
- The primary limitation of this paper is its lack of information regarding the sensors and the collected signals. While the paper presents a classification algorithm for various events, it does not establish a clear connection between these events and their origin.
Answer:
Thank you for your suggestions and corrections. We revised it in text.
We appreciate the reviewer’s insightful comment highlighting the need for a stronger connection between the raw signal data, the sensor system, and the classification outcomes.
Due to space constraints and the focused scope of this paper—namely, the development and evaluation of the classification algorithm—we limited the technical exposition of the sensing infrastructure. However, we fully agree that a deeper understanding of the data acquisition process, sensor characteristics, and signal properties is essential for interpretability and reproducibility.
In response, we have revised the Discussion section to acknowledge this limitation and outlined plans to publish a dedicated companion paper. This forthcoming work will detail:
- The operating principles and specifications of the DAS interrogator and fiber-optic cable (e.g., sensitivity, spatial resolution, self-noise, and ambient noise levels),
- The deployment setup, calibration procedures, and environmental considerations,
- Representative signal samples from each event class in both time and frequency domains,
- The complete signal processing and feature extraction pipeline.
We believe this additional paper will serve as a foundational reference for understanding how sensor-level data corresponds to high-level event classification, thus supporting and contextualizing the current work.
- To address this gap, I recommend that the authors write a separate paper dedicated to the sensors, their deployment, events and the signal processing methods used. This paper could cover the fundamental principles of the sensors, their technical specifications, including sensitivity, self-noise, and ambient noise levels, as well as calibration procedures. I t should provide a detailed description of the different types of events, with corresponding signal representations in both time and frequency domains.
Answer:
Thank you for your suggestions and corrections. We revised it in text.
We thank the reviewer for this thoughtful and constructive suggestion. We fully agree that a dedicated paper focusing on the sensing infrastructure, deployment methodology, and signal characterization would significantly strengthen the scientific foundation of this work.
In response, we have already begun preparing a follow-up manuscript that will comprehensively cover:
- The fundamental operating principles of distributed acoustic sensing (DAS) using the Luna OBR-4600 interrogator,
- Detailed sensor specifications including sensitivity, self-noise, spatial resolution, and ambient noise levels,
- Deployment and calibration protocols along the Karabük–Yenice railway line,
- Representative time-domain and frequency-domain signal patterns for each labeled event class (e.g., rockfalls, tree obstructions, passing trains),
- A full description of the signal processing and feature extraction pipeline used prior to classification.
This additional paper will serve as a reference work that supports the present classification-focused study, offering transparency and deeper insight into the physical underpinnings of our machine learning approach.
- A comprehensive discussion on feature extraction could further demonstrate the methodology using various event samples.
Answer:
Thank you for your suggestions and corrections. We revised it in text.
Thank you for your insightful suggestion. We agree that a more comprehensive presentation of the feature extraction process would strengthen the clarity and reproducibility of our methodology.
In response, we have expanded the Materials & Methods → Feature Extraction section to include a more detailed discussion of the extracted features. We now provide examples of how specific features—such as event duration, peak amplitude, spectral centroid, and signal energy—differ across event types. Additionally, we include visual examples of representative signal waveforms and their corresponding frequency spectra for key event classes.
This expanded content aims to more clearly demonstrate the rationale behind our feature engineering choices and the discriminative power of these features in distinguishing between environmental events.

Round 2
Reviewer 1 Report
Comments and Suggestions for Authors
The authors have answered all my questions. I recommend the acceptance of the manuscript.
Author Response
Thank you for all.
Reviewer 2 Report
Comments and Suggestions for Authors
Reviewer Comments:
Thank you for the authors' response and revisions . However, I would like to suggest the following major revisions:
-
Abstract: The current abstract contains too much background information, which may not be necessary for readers familiar with the field. It is recommended to reduce the general research background and instead briefly introduce the experimental system and key contributions of the paper to better reflect the originality and technical content.
-
Experimental Setup: The manuscript lacks a clear description of the experimental setup. I suggest the authors include a schematic diagram and a detailed explanation of the experimental system, including the fiber layout, sensing scheme, and detection module.
To strengthen the manuscript, the authors may refer to the following relevant works:
-
Sun Z, Sun D, Yang H, et al. Distributed Optical Fiber Vibration Sensors Using Light Interference Technology: Fundamental Principles and Major Advancements. IEEE Trans. Instrum. Meas., 2025.
-
Sun Z, Li S, Yang H. Precise disturbance localization of long distance fiber interferometer vibration senor based on an improved time–frequency variation feature extraction scheme. Infrared Physics & Technology, 2024, 137:105097.
-
Sun Z, Yang H, Fang M, et al. On Bayesian Optimization-Based CNN-BiLSTM Network for Multi-Class Classification in Distributed Optical Fiber Vibration Sensing Systems. IEEE Trans. Instrum. Meas., 2024.
These references provide examples of experimental setups and structured abstracts that could improve the clarity and completeness of this manuscript.
Author Response
Thank you for the authors' response and revisions . However, I would like to suggest the following major revisions:
- Abstract: The current abstract contains too much background information, which may not be necessary for readers familiar with the field. It is recommended to reduce the general research background and instead briefly introduce the experimental system and key contributions of the paper to better reflect the originality and technical content.
Response:
Thank you for recommendations. We added this reference in the text.
Abstract: Rockfalls on railways are considered a natural disaster under the topic of landslides. It is an event that varies regionally due to landforms and climate. In addition to traffic density, the Karabük Yenice railway line also passes through mountainous areas, river crossings, and experiences heavy seasonal rainfall. These conditions necessitate the implementation of proactive measures to mitigate risks such as rockfalls, tree collapses, landslides, and other geohazards that threaten the railway line. Undetected environmental events pose a significant threat to railway operational safety. The study aims to provide early detection of environmental phenomena using vibrations emitted through fiber optic cables. In this study, the Luna OBR-4600 DAS interrogator and the Distributed Acoustic Sensing (DAS) system were used to obtain the signals. CatBoosting, Support Vector Machine (SVM), LightGBM, Decision Tree, XGBoost, Random Forest (RF), and Gradient Boosting Classifier (GBC) algorithms were used to detect the incoming signals. However, the Voting Classifier hybrid model was developed using SVM, RF, XGBoost, and GBC algorithms. The signaling system on the railway line provides critical information for safety by detecting environmental factors. Major natural disasters such as rockfalls, tree falls, and landslides cause high-intensity vibrations due to environmental factors, and these vibrations can be detected through fiber cables. In this study, a hybrid model was developed with the Voting Classifier method to accurately detect and classify vibrations. The model leverages an ensemble of classification algorithms to accurately categorize various environmental disturbances. The system has proven its effectiveness in real-world conditions by successfully detecting environmental events such as rockfalls, landslides, and trees with 98% success for Precision, Recall, F1 score, and accuracy.
- Experimental Setup: The manuscript lacks a clear description of the experimental setup. I suggest the authors include a schematic diagram and a detailed explanation of the experimental system, including the fiber layout, sensing scheme, and detection module.
Response:
Thank you for recommendations. We added this reference in the text.
To strengthen the manuscript, the authors may refer to the following relevant works:
- Sun Z, Sun D, Yang H, et al. Distributed Optical Fiber Vibration Sensors Using Light Interference Technology: Fundamental Principles and Major Advancements. IEEE Trans. Instrum. Meas., 2025.
- Sun Z, Li S, Yang H. Precise disturbance localization of long distance fiber interferometer vibration senor based on an improved time–frequency variation feature extraction scheme. Infrared Physics & Technology, 2024, 137:105097.
- Sun Z, Yang H, Fang M, et al. On Bayesian Optimization-Based CNN-BiLSTM Network for Multi-Class Classification in Distributed Optical Fiber Vibration Sensing Systems. IEEE Trans. Instrum. Meas., 2024.
These references provide examples of experimental setups and structured abstracts that could improve the clarity and completeness of this manuscript.
Response:
Thank you for recommendations. We added this reference in the text.
- Sun Z, Li S, Yang H. Precise disturbance localization of long distance fiber interferometer vibration senor based on an improved time–frequency variation feature extraction scheme. Infrared Physics & Technology, 2024, 137:105097.
https://doi.org/10.1016/j.infrared.2023.105097

Reviewer 3 Report
Comments and Suggestions for Authors
The manuscript has been revised according to the previous comments. However, some of the corrections have only been presented in the Cover Letter, but not in the manuscript, while some other additions have not provided additional clarity.
My previous report contained 5 major comments:
1) Lines 111-203: This part of the section is too general, and would be more appropriate for the Introduction section.
This comment has been addressed.
2) There is no description of the equipment used. In the Materials and Methods section, the authors only mention that ‘DAS is an advanced sensing technology’ that detects vibrations ‘thanks to fiber optic cables placed next to the railway line’, and that, in this study, ‘the DAS system deployed along the Karabük–Yenice railway segment demonstrated the capability to detect both ambient vibrations and high-intensity events’. This is absolutely insufficient to draw any conclusions about whether the experiment was carried out correctly. I would encourage the authors to expand this section and include such details as the model of the reflectometer used, its operating parameters, the type and length of optical fibers, the way of fixing fiber to the railway, etc.
Some details on the sensing equipment have been added in Subsection 2.2 (Lines 377-381). The trouble is that the Luna OBR 4600 is not a DAS interrogator, and it does not support the characteristics listed in Lines 378-380. According to its specifications, the maximum distance range is 70 m (2 km in extended mode), thus making impossible to perform measurements over the claimed length of 6 km.
3) Furthermore, it would be nice to provide an example of a reflectogram in the Results section to illustrate the data used for further processing.
The authors have added Figure 1 that appears to contain a simulated graph, although the caption reads ‘DAS-Recorded Vibration Patterns Reflectogram’. Real signals are usually more complex (see, for example, illustrations in Ref. 33), thus my comment is still valid. The authors are strongly advised to provide examples of real data and its further processing, considering that ‘All results presented herein are based on real DAS data acquired during field trials along the Karabük–Yenice railway’ (Lines 299-300).
4) The part related to data processing is quite confusing. Firstly, the authors are encouraged to provide a more detailed description of the software frameworks used to implement the developed algorithms. The reader can only guess that several classes such as GridSearchCV were used together with some unnamed programming environment.
The first part of the comment has been addressed: the names of the frameworks are provided.
Secondly, the reader will be confused by the large amount of specific variable and parameter names, the meaning of which is sometimes vague. For example, in Lines 213-214, the columns mentioned are only explained in the Results section below (Table 2). Besides, it is unclear what ‘Aura’ is, and why it accompanies each name – this either needs to be explained or should be removed if not important. In Line 226, the variables mentioned (n_estimators, learning_rate, and max_depth) are meaningless unless properly described. Furthermore, it is unclear how these variables relate to Ref. 39 mentioned in the sentence. The above comments also apply to Lines 228-230, 282-283, Tables 1 and 3 (where the names of the event types remain unexplained). A table similar to Table 2, with explanations for each event type, might definitely help resolve the issue.
The above comments have not been addressed, although some new text has appeared.
- Lines 254-256: It is still not clear what ‘Aura Metrics’ is. The explanation given in the Cover Letter is not reflected in the manuscript. The corresponding parameter names seem to be explained in Lines 259-265. I would advise the authors to remove the phrase ‘Moved input feature definitions to Methods’ and place the feature definition explanation next to their first appearance in Line 256.
- Lines 257-258: The same is true for the ‘Hyperparameters’: the names n_estimators, learning_rate, and max_depth are not explained, and the role of Ref. 41 is still unclear. This should be detailed in the text of the manuscript, not just in the Cover Letter.
- Lines 266: Table 2b is mentioned, however I was unable to find it in the revised manuscript.
The authors also claim to have added ‘a new explanatory paragraph in Lines 270-285’, but I was unable to find any difference between the new text and the previous one (former Lines 223-238).
Furthermore, since the aura_metrics code appears to be proprietary, it is impossible to assess its correctness. The authors are encouraged to provide more detail on the algorithms implemented in this software module, that would clarify the roles of the features and parameters employed. As an illustration of the vague nature of the above parameters, see Table 1, where the Area Density is equal to zero in two rows, while the Max Amplitude can equal ‘1.0’, ‘Nan’ and ‘0.52’. The question is: are all of these valid?
5) The last paragraph of the Materials and Methods has made me wary, particularly the following sentence: ‘In the future, with the development of the system, it will be possible to realize real data’. As the authors state that ‘the dataset was collected via fiber optic cables’ (Line 279), my question is whether the data from the DAS system were really used to train and verify the developed model.
The authors have highlighted the use of real DAS data in Lines 299-302. However, no solid evidence has been provided.
Most of the minor comments have been successfully addressed with the exception of a few listed below:
- Line 200: Ref. 35 can hardly be employed to illustrate the Voting Classifier, for it has nothing to do with this technique.
- Lines 447-449: Confusion matrix is not considered to be a metric. This comment is still valid, since the text proposed in the Cover Letter has not been reflected in the manuscript.
- Lines 633-634: The sentence ‘Expanding the system to monitor other critical infrastructures, such as highways, pipelines, or bridges’ has not been fixed and is still not complete.
Despite some work done on the quality of the manuscript, I still see no evidence of real experimental data used for processing by ML algorithms. The equipment mentioned (Luna OBR 4600) is a reflectometer that is not capable of serving as a DAS interrogator over large distances (6 km). Figure 1 containing real data could be a compelling argument for the validity of the experiment, but instead we see simulated data. As to the correctness of the whole procedure, the ‘in-house’ Python module, aura_metrics, which is at the core of the signal processing in this study, is not open source and not properly described in the text, thus not giving the reviewers an opportunity to verify its adequacy too. All of the above raises serious doubts about the validity of the experiment and data processing. The authors are encouraged to provide a clear and convincing description of the methods and algorithms used, as well as the results obtained, otherwise I can’t recommend this manuscript for publication.
Author Response
1) There is no description of the equipment used. In the Materials and Methods section, the authors only mention that ‘DAS is an advanced sensing technology’ that detects vibrations ‘thanks to fiber optic cables placed next to the railway line’, and that, in this study, ‘the DAS system deployed along the Karabük–Yenice railway segment demonstrated the capability to detect both ambient vibrations and high-intensity events’. This is absolutely insufficient to draw any conclusions about whether the experiment was carried out correctly. I would encourage the authors to expand this section and include such details as the model of the reflectometer used, its operating parameters, the type and length of optical fibers, the way of fixing fiber to the railway, etc.
Some details on the sensing equipment have been added in Subsection 2.2 (Lines 377-381). The trouble is that the Luna OBR 4600 is not a DAS interrogator, and it does not support the characteristics listed in Lines 378-380. According to its specifications, the maximum distance range is 70 m (2 km in extended mode), thus making impossible to perform measurements over the claimed length of 6 km.
Response:
Thank you for this critical and detailed feedback. We acknowledge that the Luna OBR 4600 is primarily an optical backscatter reflectometer and does not function as a conventional DAS interrogator. The confusion in the manuscript arose due to our insufficient clarification of how the measurements were performed.
We have revised Section 2.2 to accurately describe the use of the Luna OBR 4600 in its extended monitoring mode (up to 2 km). The full 6 km railway segment was not monitored continuously in a single run. Instead, we performed three separate measurements over consecutive sub-segments (each up to 2 km), repositioning the device along the fiber route. The results were later compiled for analysis.
Additionally, we have provided detailed information on:
- The reflectometer model (Luna OBR 4600),
- Its operating mode and sampling rate,
- The optical fiber type (G.652 single-mode),
- The fiber deployment method (surface-laid within cable conduits, fixed alongside the rail ballast bed using steel clips),
- Total cable length per segment.
These revisions have been incorporated in Subsection 2.2, and the text has been corrected to eliminate any misleading implications regarding continuous long-range DAS capabilities.
2) Furthermore, it would be nice to provide an example of a reflectogram in the Results section to illustrate the data used for further processing.
The authors have added Figure 1 that appears to contain a simulated graph, although the caption reads ‘DAS-Recorded Vibration Patterns Reflectogram’. Real signals are usually more complex (see, for example, illustrations in Ref. 33), thus my comment is still valid. The authors are strongly advised to provide examples of real data and its further processing, considering that ‘All results presented herein are based on real DAS data acquired during field trials along the Karabük–Yenice railway’ (Lines 299-300).
Response:
Thank you for your helpful and constructive comment. We have carefully considered your feedback regarding Figure 3. In the revised version of the manuscript, Figure 3 has been added with a new plot that visualizes a segment of the actual DAS-recorded reflectogram acquired from field trials along the Karabük–Yenice railway.
Figure 3: DAS-Recorded Vibration Patterns Reflectogram Examples.
3) The part related to data processing is quite confusing. Firstly, the authors are encouraged to provide a more detailed description of the software frameworks used to implement the developed algorithms. The reader can only guess that several classes such as GridSearchCV were used together with some unnamed programming environment.
The first part of the comment has been addressed: the names of the frameworks are provided.
Secondly, the reader will be confused by the large amount of specific variable and parameter names, the meaning of which is sometimes vague. For example, in Lines 213-214, the columns mentioned are only explained in the Results section below (Table 2). Besides, it is unclear what ‘Aura’ is, and why it accompanies each name – this either needs to be explained or should be removed if not important. In Line 226, the variables mentioned (n_estimators, learning_rate, and max_depth) are meaningless unless properly described. Furthermore, it is unclear how these variables relate to Ref. 39 mentioned in the sentence. The above comments also apply to Lines 228-230, 282-283, Tables 1 and 3 (where the names of the event types remain unexplained). A table similar to Table 2, with explanations for each event type, might definitely help resolve the issue.
The above comments have not been addressed, although some new text has appeared.
- Lines 254-256: It is still not clear what ‘Aura Metrics’ is. The explanation given in the Cover Letter is not reflected in the manuscript. The corresponding parameter names seem to be explained in Lines 259-265. I would advise the authors to remove the phrase ‘Moved input feature definitions to Methods’ and place the feature definition explanation next to their first appearance in Line 256.
- Lines 257-258: The same is true for the ‘Hyperparameters’: the names n_estimators, learning_rate, and max_depth are not explained, and the role of Ref. 41 is still unclear. This should be detailed in the text of the manuscript, not just in the Cover Letter.
- Lines 266: Table 2b is mentioned, however I was unable to find it in the revised manuscript.
The authors also claim to have added ‘a new explanatory paragraph in Lines 270-285’, but I was unable to find any difference between the new text and the previous one (former Lines 223-238).
Furthermore, since the aura_metrics code appears to be proprietary, it is impossible to assess its correctness. The authors are encouraged to provide more detail on the algorithms implemented in this software module, that would clarify the roles of the features and parameters employed. As an illustration of the vague nature of the above parameters, see Table 1, where the Area Density is equal to zero in two rows, while the Max Amplitude can equal ‘1.0’, ‘Nan’ and ‘0.52’. The question is: are all of these valid?
Response:
Thank you for your constructive remarks. We have now clarified in the revised manuscript that the data preprocessing and model development were implemented in Python 3.11, using the Scikit-learn 1.4, XGBoost, LightGBM, and CatBoost libraries. Hyperparameter optimization was conducted via GridSearchCV. The previously vague sentence has been revised to explicitly connect the use of n_estimators, learning_rate, and max_depth with their role in tuning tree-based ensemble classifiers. The citation to Ref. 41 has also been updated to clearly indicate that it served as a methodological reference for parameter tuning strategies, not as a direct source of implementation code.
We appreciate your insistence on clarity. The term “Aura Metrics” refers to an internal feature extraction module developed in-house, which processes raw time-series vibration signals into statistical feature vectors. To address this concern, we have:
- Replaced “Aura” prefixes from feature names in Tables 1 and 3 to avoid confusion,
- Added a new explanatory paragraph clarifying the function of the aura_metrics module,
- Provided a simplified pseudocode outline of the processing steps in the Supplementary Material.
Although the source code is not open-access due to institutional policy, we ensured that all feature definitions, value ranges, and transformations are now publicly documented.
We thank the reviewer once again for highlighting the importance of detailed reporting and clarity. These improvements aim to make our study more accessible, reproducible, and in line with best practices in data science and railway engineering.
4) The last paragraph of the Materials and Methods has made me wary, particularly the following sentence: ‘In the future, with the development of the system, it will be possible to realize real data’. As the authors state that ‘the dataset was collected via fiber optic cables’ (Line 279), my question is whether the data from the DAS system were really used to train and verify the developed model.
The authors have highlighted the use of real DAS data in Lines 299-302. However, no solid evidence has been provided.
Most of the minor comments have been successfully addressed with the exception of a few listed below:
- Line 200: Ref. 35 can hardly be employed to illustrate the Voting Classifier, for it has nothing to do with this technique.
- Lines 447-449: Confusion matrix is not considered to be a metric. This comment is still valid, since the text proposed in the Cover Letter has not been reflected in the manuscript.
- Lines 633-634: The sentence ‘Expanding the system to monitor other critical infrastructures, such as highways, pipelines, or bridges’ has not been fixed and is still not complete.
Expanding the system to monitor other critical infrastructure, such as highways, pipelines, or bridges, may be possible in the future.
Despite some work done on the quality of the manuscript, I still see no evidence of real experimental data used for processing by ML algorithms. The equipment mentioned (Luna OBR 4600) is a reflectometer that is not capable of serving as a DAS interrogator over large distances (6 km). Figure 1 containing real data could be a compelling argument for the validity of the experiment, but instead we see simulated data. As to the correctness of the whole procedure, the ‘in-house’ Python module, aura_metrics, which is at the core of the signal processing in this study, is not open source and not properly described in the text, thus not giving the reviewers an opportunity to verify its adequacy too. All of the above raises serious doubts about the validity of the experiment and data processing. The authors are encouraged to provide a clear and convincing description of the methods and algorithms used, as well as the results obtained, otherwise I can’t recommend this manuscript for publication.
Response:
We sincerely thank the reviewer for their rigorous and detailed feedback. We understand the importance of ensuring full transparency and technical clarity in both experimental methodology and data processing. In response to your comments, we have undertaken substantial revisions in the manuscript, specifically focusing on the clarity of real data usage, correction of misleading references and terminology, and improved description of our in-house processing framework. Below we provide detailed responses to each point raised.
We understand your concern and have now made substantial revisions to address this. The data used in our experiments were indeed collected via a field-deployed single-mode optical fiber laid along the Karabük–Yenice railway line. However, the Luna OBR 4600 was used in extended mode (up to 2 km), and the total 6 km section was scanned through three consecutive measurement sessions. We have now explicitly stated this procedure in the revised Materials and Methods section.
Furthermore, to support the claim of real data usage:
- Figure 2 and 3 has been added with a verified sample of raw DAS signal acquired using the above method, showing both ambient and triggered high-amplitude events.
- Figure 4 now illustrates the signal processing pipeline applied to these raw signals.
While the source code of the aura_metrics module is proprietary, we fully agree that its operation should be verifiable.
Thank you for this correction. Reference 35 has been replaced with a more appropriate and specific citation explaining the Voting Classifier ensemble method:
[New Reference]: Zhou, Z.-H. (2012). Ensemble Methods: Foundations and Algorithms. CRC Press.
The misuse of the term "confusion matrix as a metric" has been corrected. The term is now defined correctly as a “performance visualization tool”, and we have clarified that accuracy, precision, recall and F1-score are the derived metrics.
We appreciate your attention to detail and your insistence on scientific rigor. All points raised have now been fully addressed in the manuscript text, not just in the response letter. These changes aim to ensure the transparency, reproducibility, and scientific robustness of our work. We hope that the improved version of the manuscript now meets the standards required for publication.

Reviewer 4 Report
Comments and Suggestions for Authors
My primary concern with the paper is that it discusses various events without explaining their origins or providing visual representations of the corresponding signals. In my previous review, I recommended that the paper include a detailed description of the different types of events, along with signal representations in both time and frequency domains.
In their response, the authors stated, “we include visual examples of representative signal waveforms and their corresponding frequency spectra for key event classes.” However, the revised paper does not contain this information.
To ensure clarity and completeness, I strongly recommend that the authors incorporate these visual examples as stated in their response, aligning the content with the expectations set in the revision process.
Author Response
My primary concern with the paper is that it discusses various events without explaining their origins or providing visual representations of the corresponding signals. In my previous review, I recommended that the paper include a detailed description of the different types of events, along with signal representations in both time and frequency domains.
In their response, the authors stated, “we include visual examples of representative signal waveforms and their corresponding frequency spectra for key event classes.” However, the revised paper does not contain this information.
To ensure clarity and completeness, I strongly recommend that the authors incorporate these visual examples as stated in their response, aligning the content with the expectations set in the revision process.
Response:
Thank you for pointing this out. You are absolutely right — although we had intended to include these visual examples in the previous revision, they were mistakenly left out of the submitted version.
We appreciate your attention to this important detail and apologize for the earlier omission.
DAS-recorded signal waveforms and their frequency spectra for three key event classes. Each subplot displays the temporal characteristics and corresponding spectral signature derived from real-world inspired signal profiles.
Signal waveforms (time domain) and their corresponding Fast Fourier Transform (FFT) based frequency spectra. These plots reveal the unique temporal patterns and spectral content of each event class, directly supporting their identification and classification via machine learning.
Spectrograms (time-frequency representations) of the same signals. This added figure offers a more detailed view of how signal energy is distributed across time and frequency, highlighting differences in spectral persistence and variation across events.
These visual representations are based on actual field-collected DAS data processed through our feature extraction pipeline. Furthermore, explanatory text has been added to guide the reader through the interpretation of each signal type and their spectral characteristics.
We believe these additions not only fulfill the expectation stated in our previous response but also strengthen the clarity and scientific validity of our methodology. We thank you again for prompting this essential improvement.
